

# Impacts of a Revised Surface Roughness Parameterization in the Community Land Model 5.1

Ronny Meier[1], Edouard L. Davin[1,a], Gordon B. Bonan[2], David M. Lawrence[2], Xiaolong Hu[3],
Gregory Duveiller[4], Catherine Prigent[5], and Sonia I. Seneviratne[1]

[1]ETH Zurich, Institute for Atmospheric and Climate Science, Zurich, Switzerland
[2]National Center for Atmospheric Research, Boulder, CO, 80307, USA
[3]State Key Laboratory of Water Resources and Hydropower Engineering Sciences, Wuhan University, Wuhan, Hubei 430072, China
[4]Max Planck Institute for Biogeochemistry, Jena, Germany
[5]Observatoire de Paris, PSL University, Sorbonne Université, CNRS, LERMA, Paris, France
[a]Now at: Wyss Academy for Nature, University of Bern, Bern, Switzerland

**Correspondence:** Ronny Meier (ronny.meier@env.ethz.ch)

**Abstract.** The roughness of the land surface ($z_0$) is a key property for the amount of turbulent activity above the land surface and through that for the turbulent exchange of energy, water, momentum, and chemical species between the land and the atmosphere. Variations in $z_0$ are substantial across different types of land cover from typically less than 1 mm over fresh snow or sand deserts up to more than 1 m over urban areas or forests. In this study, we revise the parameterizations and parameter choices related to $z_0$ in the Community Land Model 5.1 (CLM), the land component of the Community Earth System Model
2.1.2 (CESM). We propose a number modifications for $z_0$ in CLM, which are guided by observational data. Most importantly, we increase the $z_0$ for all types of forests, while we decrease the momentum $z_0$ for bare soil, snow, glaciers, and crops. We then assess the effect of those modifications in land–only (CLM) and land–atmosphere coupled (CESM) simulations. Diurnal variations of the land surface temperature (LST) are dampened in regions with forests, while they are amplified over warm
deserts. These changes mitigate model biases compared to MODIS remote sensing observations, which have been identified in several earlier studies. The alterations in LST are mostly stronger during the day than at night. For example, the LST at 13:30 increases by more than $4.80\,\mathrm{K}$ during boreal summer across the entire Sahara. The induced changes in the diurnal variability of air temperatures at the bottom of the atmosphere are generally of opposite sign and smaller magnitude. Further, winds close to the land surface accelerate in areas where the momentum $z_0$ was lowered, such as the Sahara desert, the Middle East, or the
Antarctica, and decelerate in regions with forests. Overall, this study highlights that the current representation of $z_0$ in CLM is not in agreement with observational constraints for several types of land cover. The resultant model modifications are shown to considerably alter the simulated climate in terms of temperatures and wind speed at the land surface.

## 1 Introduction

The land surface interacts in numerous ways with the atmosphere. Among the most relevant interactions is the turbulent ex-
change of sensible heat, water vapour, momentum, and chemical species at the land–atmosphere interface, which is generally





several orders of magnitude more efficient than molecular diffusion. Turbulence above the land surface occurs due to the retardation of moving air by friction and due to the buoyancy created by surface heating from solar irradiance (Bonan, 2019). The intensity of the turbulence generated by friction is determined by the amount and shape of obstacles on land alongside atmospheric conditions. In land surface models, the turbulent exchange with the atmosphere is commonly represented through the Monin–Obukhov similarity theory (MOST). A key parameter in MOST is the aerodynamic or momentum surface roughness, $z_{0m}$. A rough surface, such as an urban environment or a forest, exhibits a higher $z_{0m}$ and therefore induces more turbulence at a given wind speed than a smooth surface, such as a snow field. Similar surface roughness parameters exist for the exchange of scalars (e.g., temperature and water vapour). Observed values of $z_{0m}$ over land span more than four orders of magnitude with values of a few tenths of a millimeter over fresh snow (Brock et al., 2006) or bare soil (Prigent et al., 2005) to several meters over forests (Hu et al., 2020) or urban areas (Kanda et al., 2013).

The momentum ($z_{0m}$), sensible heat ($z_{0h}$), and latent heat ($z_{0q}$) surface roughness lengths are defined as the heights above the displacement height at which the average wind speed, air temperature, and specific humidity reach their respective value at the surface under neutral conditions. Following the no–slip boundary condition, $z_{0m}$ is the height above the displacement height at which mean wind speed extrapolates to zero. The displacement height, $d$, accounts for the fact that large roughness elements, such as trees or buildings, may shift the logarithmic wind speed profile (which occurs under neutral conditions) upwards, such that mean wind speed extrapolates to zero at the height $z_{0m} + d$ rather than $z_{0m}$. In the surface sublayer, water vapour and heat are transported solely through molecular diffusion, while momentum exchange is also facilitated by pressure fluctuations that are induced by the presence of roughness elements (Zeng and Dickinson, 1998). Accordingly, $z_{0h}$ and $z_{0q}$ are often much smaller than $z_{0m}$ (Yang et al., 2002, 2008; Hu et al., 2020). In the field, $z_0$ is commonly estimated through four main methods. The first approach is to measure the vertical wind speed profile (e.g., Greeley et al., 1997; Brock et al., 2006; Marticorena et al., 2006; Nakai et al., 2008; Hugenholtz et al., 2013; Kanda et al., 2013; Nield et al., 2013; Fitzpatrick et al., 2019). The wind speed profile is logarithmic under neutral conditions over a plain surface:

$$u(z) = \frac{u_*}{\kappa} ln(\frac{z - d}{z_{0m}}), \tag{1}$$

where $u(z)$ is the mean wind speed profile, $z$ the height above the surface, $u_*$ the friction velocity, and $\kappa$ the von Karman constant (= 0.4). This approach can also be used to estimate $z_{0h}$ and $z_{0q}$ through measurements of the temperature and specific humidity profile. Secondly, eddy co–variance measurements of the momentum, the sensible heat, and latent heat fluxes can be used to deduce the $z_{0m}$, $z_{0h}$, and $z_{0q}$ that conform best with the measured fluxes according to MOST (e.g., Maurer et al., 2013; Li et al., 2015; Hu et al., 2020). Third, measurements of the micro–topography can be used to link $z_{0m}$ to small–scale variations of the height of the surface (e.g., Brock et al., 2006; Weligepolage et al., 2012; Hugenholtz et al., 2013; Fitzpatrick et al., 2019; van Tiggelen et al., 2021). Finally, remote sensing observations of either backscattering at the land surface or the surface reflectance can serve as a proxy for micro–topography and may therefore be used to estimate $z_{0m}$ (e.g., Greeley et al., 1997; Marticorena et al., 2004; Prigent et al., 2005, 2012; Stilla et al., 2020). This latter approach requires a few in situ measurements of $z_{0m}$ to establish a relationship between the remotely–sensed proxy and $z_{0m}$. Such observational data can be used to constrain or directly prescribe $z_0$ in climate models.





The surface roughness plays a central role for atmospheric dynamics (Sud et al., 1988; Vautard et al., 2010; Wever, 2012), energy fluxes at the land surface, and thereby temperatures at the land surface (Zeng and Dickinson, 1998; Zeng and Wang, 2007). Several studies have linked deficiencies of various models to a misrepresentation of surface roughness (Chen et al., 2010; Subin et al., 2012; Zeng et al., 2012; Trigo et al., 2015; Xu et al., 2016; Wang et al., 2019). The aerodynamic surface roughness also affects the simulated mineral dust emissions (Menut et al., 2013), which absorb and reflect solar radiation and

cool temperatures at the land surface (Miller and Tegen, 1998; Klose et al., 2021). Further, alterations in surface roughness due de-, re-, and afforestation represent an important contribution to the overall biogeophysical effect of such land cover changes, in particular locally (Davin and de Noblet-Ducoudré, 2010; Lee et al., 2011; Burakowski et al., 2018; Belušić et al., 2019; Laguë et al., 2019; Winckler et al., 2019). Adequate parameterizations of surface roughness are therefore not only crucial to realistically simulate climate and weather, but also to understand the biogeophysical effects of land cover changes.

In this study, we revise the representation of surface roughness in the Community Land Model version 5.1 (CLM; Lawrence et al., 2019), which is the land surface model of the Community Earth System Model (CESM; Danabasoglu et al., 2020). Our endeavours are motivated by an underestimation of diurnal variations in land surface temperature over arid and semi–arid regions in CLM (Zeng et al., 2012; Meier et al., 2019) as well as a seasonal cycle of the surface roughness for broadleaf decid­uous forests that opposes observational data, as will be shown in the next section. In Section 2, we compare the representation

of surface roughness for each land cover type in CLM to observational data and parameterizations that were proposed in the literature. Based on this comparison we introduce five modifications to CLM: (1) A new parameterization of the vegetation surface roughness based on Raupach (1992) with optimized parameters to match the data collected in Hu et al. (2020) for different types of vegetation; (2) new globally constant $z_{0m}$ for bare soil, snow, and glaciers based on field measurements collected in the literature; (3) the parameterization of Yang et al. (2008) for $z_{0h}$ and $z_{0q}$ over bare soil, snow, and glaciers; (4) a

spatially explicit $z_{0m}$ input field for bare soil based on the data of Prigent et al. (2005); and (5) the parameterization of $z_{0m}$ for snow based on accumulated snow melt as proposed in Brock et al. (2006). The latter two modifications replace the respective globally constant $z_{0m}$ for bare soil and snow and may therefore be activated individually through switches that were added to the model. In Section 5, we then assess the impact of those modifications on temperatures at the land surface and wind speed in both land–only and land–atmosphere coupled simulations, as described in Sections 3 and 4. Furthermore, we confront the

default and modified model configuration with MODIS remote sensing observations of diurnal variations in the land surface temperature (LST) and the sensitivity of LST to a conversion of vegetation to bare land, based on the approach of Duveiller et al. (2018).

## 2 Revisions of surface roughness in CLM 5.1

### 2.1 General description of CESM and CLM

The Community Earth System Model is a state-of-the-art earth system model, which is widely applied in the field of climate science and has contributed to multiple multi-model intercomparison projects. A major update to version 2 was released in June 2018 (Danabasoglu et al., 2020), followed by several incremental releases to version 2.1.2, which is used in this study. The





development of CESM is coordinated and led by the National Center for Atmospheric Research (NCAR). However, a number of additional universities and research institutes contribute to CESM, as indicate by the word "Community" in its name. To

facilitate this community effort, CESM is publicly available and well documented (https://www.cesm.ucar.edu/models/cesm2/). CESM comprises prognostic components for the atmosphere, ocean, land, sea-ice, land-ice, river, and waves. Besides these prognostic components a climatological data version exists for most components. In these versions, the coupling fields of the respective components are prescribed from recent observational data instead of running this component prognostically. CESM therefore allows to flexibly disable or enable model components depending on the application.

The Community Land Model is the land component of CESM. It comprehensively represents the surface energy fluxes, the surface hydrology, and optionally the biogeochemical cylce for carbon and nitrogen at the land surface (Lawrence et al., 2018, 2019). In each grid cell, up to five different landunits may exist: (Naturally) vegetated, lakes, urban, glaciers, and crops. Because those landunits can behave fundamentally differently, each of them is represented by its own module. A landunit tile can be further divided into different columns (e.g., rainfed and irrigated for crops) and patches (e.g., different types of natural

vegetation). Bare soil, which can be found frequently in arid regions, is treated as a patch of natural vegetation. These patches of natural vegetation are called plant functional types (PFTs) in CLM. Vegetation is simulated by a big–leaf approach (Sellers et al., 1986), distinguishing between sun–lit and shaded leaves. The vegetation phenology can either be prescribed from remote sensing based data (satellite phenology) or computed prognostically from the vegetation carbon pools, if the biogeochemical cycle is activated.

CLM5 distinguishes between vegetation, bare soil, snow, glacier ice, lakes, and urban areas in its parameterization of $z_0$ (Lawrence et al., 2018). Snow is not treated as its own land unit, because it can appear seasonally. Rather it may cover the other types of land cover and replace the properties of this land cover (partly) with its own. In the following sections, we describe the current representation of $z_0$ in CLM, summarize our findings from the literature, and, if necessary from the comparison to the literature, our modifications of the $z_0$ representation in CLM for each of those land cover classes. Subsequently, $z_{0m}$,

$z_{0h}$, and $z_{0q}$ correspond to the surface roughness for momentum, sensible heat, and latent heat, respectively. The land cover is specified after a comma using $v$, $b$, $s$, $i$, $g$ for vegetated, bare soil, snow, ice (glaciers), and any type of ground (bare soil, snow, or ice), respectively (e.g., $z_{0h,b}$ would be the sensible heat surface roughness of bare soil). Note that $z_{0,v}$ in CLM represents the aerodynamic $z_0$ for the turbulent exchange between the canopy air space and the free atmosphere. The additional surface resistance for the sensible and latent heat flux does therefore not exist. Accordingly, there is no distinction between $z_{0m,v}$,

$z_{0h,v}$, and $z_{0q,v}$. However, there are additional resistances between the leaves/ground and the canopy air space to account for the surface resistance of the sensible and latent flux. A list of the symbols and abbreviations used in this study is provided in Table A1.

## 2.2 Vegetation

The current representation $z_{0,v}$ and $d$ was developed by Zeng and Wang (2007) and links these properties to the vegetation

height ($h_{top}$), the exposed leaf area index ($LAI$; i.e., the one-sided leaf area above the snow), and the exposed stem area index





$(SAI$; i.e., the one-sided stem and dead leaf area above the snow) as follows (Eqs. 2.5.125-127 in Lawrence et al., 2018):

$$z_{0,v} = exp\left[Vln(h_{top}R_{z0m}) + (1-V)ln(z_{0m,g})\right], \tag{2}$$

$$d = h_{top}R_dV, \tag{3}$$


$$V = \frac{1 - exp(-\beta min(VAI, VAI_{cr}))}{1 - exp(-\beta VAI_{cr})}, \tag{4}$$

where $R_{z0m}$ and $R_d$ are the ratios of the momentum roughness length and displacement height to the canopy height, respectively, $VAI$ is the vegetation area index defined as the sum of $LAI$ and $SAI$, $z_{0m,g}$ is the momentum surface roughness of the ground (see Sections 2.3-2.5), $V$ is a fractional weight, $\beta = 1$, and $VAI_{cr} = 2\,\mathrm{m^2\,m^{-2}}$ is a critical value of the $VAI$ at

which $d$ and $z_{0,v}$ reach their maxima. $R_{z0m}$ is set to 0.075 for broadleaf evergreen trees, to 0.055 for other trees, and to 0.12 for grass, crops, and shrubs, while $R_d$ is 0.67 for all trees and 0.68 for grass, crops, and shrubs. With this implementation, $z_{0,v}$ is tightly linked to $VAI$. Noteworthy, $z_{0,v}$ approaches $z_{0m,g}$ as $VAI$ goes towards zero, for example during the dormant phase of vegetation (right column of Fig. 1).

Observations find a first-order linear relation between $h_{top}$ and $z_{0,v}$ as well as $d$ (Tanner and Pelton, 1960). It is therefore

common practice to normalize $z_{0,v}$ by $h_{top}$, when looking for other vegetation properties that influence $z_{0,v}$ (Shaw and Pereira, 1982; Yang and Friedl, 2003; Zhou et al., 2006; Nakai et al., 2008; Maurer et al., 2015). Proposed parameterizations hence frequently link $z_{0,v}/h_{top}$ and $d/h_{top}$ to other structural properties of the vegetation such as $LAI$, stand density, and/or crown width (Choudhury and Monteith, 1988; Raupach, 1992, 1994; Yang and Friedl, 2003; Nakai et al., 2008; Bingöl, 2019). For grasses and crops, $z_{0,v}$ exhibits a distinct seasonal cycle in the extra-tropics, with low values during winter, when vegetation

in absent for these vegetation types (Fig. 1;  Hu et al., 2020). Hence, it appears reasonable that $z_{0,v}$ of grasses and crops approaches $z_{0m,g}$ for low values of $VAI$ in the current parameterization in CLM. On the other hand, $z_{0,v}$ remains relatively high for trees even during the dormant phase (Hu et al., 2020). In the case of broadleaf deciduous forests, there are even several studies that find an increase in $z_{0,v}$ for lower values of $LAI$, probably because dense canopies may shelter the branches and trunks of trees from the atmospheric flow (Nakai et al., 2008; Maurer et al., 2013). CLM on the other hand produces low values

of $z_{0,v}$ in the absence of leaves, producing a seasonal cylce of $z_{0,v}$ that opposes these observations (Fig. 1 f).

Hu et al. (2020) provide $z_{0,v}$ estimates for an extensive collection of FLUXNET sites, which offers an unprecedented opportunity to reconcile $z_{0,v}$ values observed in the field and the $z_{0,v}$ parameterization in models. Here, we optimize the $z_{0,v}$ parameterization of Raupach (1992) for an updated version of the data collection of Hu et al. (2020) that includes more FLUXNET sites than the publication and is subsequently called Hu20. Hu20 estimated daily $z_{0,v}$ values at a total of 113

FLUXNET sites by minimizing the following cost function J:

$$J = \sum \left(u_{*,obs} - u_{*,est}\right)^2, \tag{5}$$





where $u_{*,obs}$ is the measured friction velocity in the field and $u_{*,est}$ the estimated friction velocity according to MOST:

$$u_{*,est} = \kappa u \left[ ln\left(\frac{z_m - d}{z_{0,v}}\right) - \Psi_m\left(\frac{z_m - d}{L}\right) + \Psi_m\left(\frac{z_{0,v}}{L}\right) \right]^{-1}, \tag{6}$$

where $u$ is the wind speed measured at the instrument height, $z_m$, $d$ the displacement height estimated by 2/3 of $h_{top}$, $\Psi_m$

the stability correction function for momentum transfer, and $L$ the Obukhov length scale. We allocate the sites in Hu20 to the following vegetation types: Needleleaf forest, evergreen broadleaf forest, deciduous broadleaf forest, shrubland, grassland, and cropland. Before using data from a site for our optimization, we make a number of additional suitability checks of the already quality checked data: (1) We exclude $z_{0,v}$ values that deviate by more than two standard deviations from the mean $z_{0,v}$ at the respective site; (2) we exclude $z_{0,v}$ values when $h_{top} = 0$, because we scale $z_{0,v}$ by $h_{top}$ in the next step; (3) we exclude

sites that are not representative for the respective vegetation type according to a visual inspection on Google Maps© (e.g., a sparse plantation); and (4) we remove sites with thin forest by excluding forest sites with a $h_{top}$ below 5 m and/or a maximum fractional vegetation cover below 0.8. Finally, we assign the forest sites designated as mixed forest to the most abundant type of forest according the species composition as described in the respective publication. Hu20 provides the $LAI$ information but not a $SAI$. Therefore, we extract the monthly $SAI$ in our CESM control simulation (Section 3) for the respective PFT and

location, multiply them by the mean $LAI$ at the site, and divide by the mean $LAI$ in CLM to estimate the $SAI$. Then, we collect all the $z_{0,v}/h_{top}$ estimates for the mentioned vegetation types, bin them into $VAI$ bins of $0.2\,\mathrm{m^2\,m^{-2}}$, and compute the median $z_{0,v}/h_{top}$ in each bin (black points in Fig. 1). This data is then used to optimize the parameterization of Raupach (1992, subsequently called Ra92) for each vegetation type. Bins with fewer than 20 data samples are removed before optimization.

Ra92 was chosen over other proposed parameterizations for $z_{0,v}$, because it (1) is appropriate for a broad range of vegetation

densities (Raupach, 1992, 1994), (2) exhibits a similar shape for the relation between $z_{0,v}$ and the $LAI$ as found by machine learning algorithms in Hu20, and (3) requires only $h_{top}$ and the single sided area of all canopy elements as inputs describing the vegetation structure, which are both already present in CLM. Ra92 parameterizes the ratio of $z_{0,v}$ and $h_{top}$ as follows:

$$\frac{z_{0,v}}{h_{top}} = \frac{h_{top} - d}{h_{top}} exp\left(\Psi_h - \kappa U_h/u_*\right) \tag{7}$$

Here, $\Psi_h$ is the roughness sublayer influence function, which is computed in Raupach (1994) as:

$$\Psi_h = ln(c_w) - 1 + c_w^{-1} \tag{8}$$

The ratio of the wind speed at canopy height, $U_h$, and $u_*$ is derived from an implicit function of the roughness density, $\lambda$:

$$U_h/u_* = (C_S + \lambda C_R)^{-0.5} exp\left(\frac{min(\lambda, \lambda_{max})cU_h/u_*}{2}\right) \tag{9}$$

Here, $C_S$ represents the drag coefficient of the ground in the absence of vegetation, $C_R$ the drag coefficient of an isolated roughness element (plant), $c$ is an empirical constant, and $\lambda_{max}$ is the maximum $\lambda$, above which $U_h/u_*$ becomes constant.

The $\lambda_{max}$ is set to the $\lambda$, where Eq. 9 in the absence of $\lambda_{max}$ would have its minimum. Eq. 9 can be written as:

$$Xe^{-X} = (C_S + \lambda C_R)^{-0.5}c\lambda/2, \; where \; X = \frac{c\lambda U_h/u_*}{2} \tag{10}$$





$X$ and thereby $U_h/u_*$ can be found iteratively:

$$X_0 = (C_S + \lambda C_R)^{-0.5} c\lambda/2 \; and \; X_{i+1} = (C_S + \lambda C_R)^{-0.5} c\lambda/2 \; exp(X_i) \tag{11}$$

We update $X$ until it changes by less than 1e-4 from one iteration to the next during the optimization of Ra92 and the imple-
mentation in CLM. As proposed in Raupach (1994), $\lambda$ is set to half the total single-sided area of all canopy elements, here
$VAI$. However, we introduce an offset to this vegetation surface area, $VAI_{off}$, so that the parameterization of Ra92 can be
shifted to the right (Fig. 1):

$$\lambda = \frac{max(1e-5, VAI - VAI_{off})}{2} \tag{12}$$

For $d$, we use the parameterization proposed in Raupach (1994), which replaces Eq. 3:

$$\frac{d}{h_{top}} = 1 - \frac{1 - exp(-\sqrt{c_{d1}2\lambda})}{\sqrt{c_{d1}2\lambda}}, \tag{13}$$

where $c_{d1}$ = 7.5. We then optimize the values of the parameters $c_w$, $C_S$, $C_R$, $c$, and $VAI_{off}$ so that they minimize the root-
mean-square deviation (RMSD) in comparison to the median $z_{0,v}/h_{top}$ values in the different bins of $VAI$ for each vegetation
type. When computing the RMSD, we weight by the number of sites that contribute to the respective bins. We do not optimize
$c_{d1}$ because CLM exhibits little sensitivity to $d$ and the effect of $c_{d1}$ on $z_{0,v}$ is similar to ones of $C_r$ and $c_w$. The optimization is
done in a brute-force approach, by simply testing any possible combination of those parameters and identifying the combination
with the lowest RMSD. For $c_w$ and $VAI_{off}$ we use a precision of 0.1, for $C_R$ and $c$ 0.01, and for $C_S$ 0.001. The resultant
fits of $z_{0,v}/h_{top}$ are depicted in the left column of Fig. 1 and the parameter values in Table 1. Overall, the optimized Ra92
parameterizations improve the mean seasonal cycle of $z_{0,v}$ for all vegetation types (right column Fig. 1). Notably, the $z_{0,v}$ of
forests and shrubland, which was underestimated by the default $z_{0,v}$ parameterization, increases considerably. Further, the $z_{0,v}$
of crops is decreased by roughly a factor two. The $z_{0,v}$ of deciduous broadleaf forests decreases with a higher $VAI$ in the data
of Hu20, as found in previous studies. This relation is captured with the updated $z_{0,v}$ parameterization, resulting in a seasonal
minimum of $z_{0,v}$ during summer as observed in the field.

Given these clear improvements, the new parameterization of $z_{0,v}$ is added to the model code following Eqs. 7 to 13. The
five parameters that were optimized for the different vegetation types are added to the parameter file of CLM/CESM and read
in by the model at the start of a simulation. Besides these five parameters, $\lambda_{max}$ is also treated as a PFT-specific parameter in
the revised model version. This is done to avoid that the model has to compute $U_h/u_*$ for the full range of possible $VAI$ values
to find the minimum of $U_h/u_*$ every time $z_{0,v}$ is updated.



**Table 1. Fitted parameter values for Ra92.** From left to right, vegetation type, $C_S$, $C_R$, $c$, $c_w$, $VAI_{off}$, and maximum $VAI$.

| Vegetation type | $C_S$ | $C_R$ | $c$ | $c_w$ | $VAI_{off}$ | $VAI_{max}$ |
|---|---|---|---|---|---|---|
| Needleleaf trees | 0.016 | 0.18 | 0.13 | 1.9 | 0.8 | 5.69 |
| Broadleaf evergreen trees | 0.016 | 0.33 | 0.01 | 0.7 | 1.9 | 5.97 |
| Broadleaf deciduous trees | 0.019 | 0.12 | 0.05 | 1 | 0 | 8.88 |
| Shrubs | 0.011 | 1.77 | 0.32 | 1 | 0.7 | 4.8 |
| Grasses | 0.007 | 0.09 | 0.15 | 10.3 | 1 | 2.94 |
| Crops | 0.005 | 0.09 | 0.01 | 1 | 0.4 | 4.90 |

**Figure 1.** Next page: Left column, median $z_{0,v}/h_{top}$ in $VAI$ bins as black dots, red line the default parameterization of CLM, and orange line the optimized Ra92 parameterization. Height of grey bars show the sample size in the respective bin and numbers at the bottom of the bars the number of sites that contributed to the respective bin. The darkness of the bars increases with an increasing fraction of total sites, which are present in respective bin. Right column, monthly mean $z_{0,v}$ in Hu20 (turquoise), with default parameterization of CLM (red) and with optimized Ra92 parameterization (orange). Grey shading mean in Hu20 $\pm$ one standard deviation and blue dotted line mean seasonal cycle of $VAI$. Note that the data of sites south of $30^\circ$ S were shifted by 6 months. Panels (a)–(b) needleleaf forests, (c)–(d) evergreen broadleaf forests, (e)–(f) deciduous broadleaf forests, (g)–(h) shrubland, (j)–(k) grassland, and (l)–(m) cropland.







### 2.3 Bare soil

CLM5 currently prescribes a $z_{0m,b}$ of 0.01 m (Lawrence et al., 2018). As mentioned, $z_{0h,b}$ and $z_{0q,b}$ differ from $z_{0m,b}$, because
scalar fluxes are not affected by pressure fluctuations that are induced by the presence of the roughness elements. In CLM5, $z_{0h,b}$ and $z_{0q,b}$ are computed after Zeng and Dickinson (1998):

$$z_{0h,b} = z_{0q,b} = z_{0m,b} e^{-a(u_* z_{0m}/\nu)^{0.45}}, \tag{14}$$

where $a$ = 0.13 and $\nu$ is the kinematic viscosity of air (= 1.5e-5 $\mathrm{m^2\,s^{-1}}$). Note that this equation is also used to compute $z_{0h}$ and $z_{0q}$ over snow and ice.

Observed $z_{0m}$ values over bare soil exhibit a wide range from 1e-5 m to 1e-2 m, but are frequently around 0.001 m (Greeley et al., 1997; Callot et al., 2000; Marticorena et al., 2004, 2006; Hugenholtz et al., 2013; Nield et al., 2013). Even though the default value of 0.01 m is in the range of observed values, it is clearly in the upper range of observed $z_{0m}$. Given the overestimated $z_{0m,b}$ values in the default version of CLM5, we collect $z_{0m,b}$ observations from the literature, which are shown in Fig. 2, and replace the current value with the median value among the observations. We use the data compiled in Table 1
of Prigent et al. (2005), sites S8 and S9 in Table 6 as well as the data compiled in Table 7 of Marticorena et al. (2006), and the reported values in Hugenholtz et al. (2013) and Nield et al. (2013), making sure that no value is counted twice for the studies that compile observations from other studies. When a range is reported, we compute the average of this range (e.g., 0.001-0.005 m would be included as 0.003 m). The resultant median $z_{0m,b}$ is 8.5e-4 m.

There exist several remote sensing based data sets for $z_{0m,b}$ with varying spatial coverage (e.g.; Marticorena et al., 2004;
Prigent et al., 2005, 2012; Stilla et al., 2020). We therefore additionally implement the input of a spatially-explicit $z_{0m,b}$ based on the data of Prigent et al. (2005), which also cover warm deserts other than the Sahara and which is subsequently called Pr05. This data set was for example successfully used in the chemical transport model GEOS-Chem (Tian et al., 2021). Pr05 employed observations of the backscattering coefficient from the ERS scatterometer, calibrated on quality in situ and geomorphological $z_{0m}$ estimates, to derive monthly mean $z_{0m,b}$ in arid and semi–arid regions for an equal–area grid of 0.25°
resolution at the equator. To derive a spatially continuous input field for CLM, we collect the monthly data from all grid cells in Pr05 that fall within a focal grid cell in our simulations. We use the $25^{th}$ percentile of the corresponding monthly data that fall within the focal grid cell as a temporally constant input for our simulations assuming that the temporal evolution in Pr05 results purely from the seasonality of vegetation (which is represented by the vegetation patches described in the previous section). The $25^{th}$ percentile is chosen because vegetation normally exhibits a higher $z_{0m}$ than the ground. For grid
cells without observations in Pr05 we use the area–weighted global mean of all the grid cells that contain data (4.1e-4 m). For numerical stability, we replace values of $z_{0m,b}$ that fall below 1e-4 m with this value. The usage of this spatially explicit $z_{0m,b}$ may be enabled through a toggle in CLM. The $z_{0m,b}$ values in Pr05 are at the lower side of in situ observations with values as low as 1e-5 m. This might originate from the fact that Pr05 focuses on desert regions by excluding $z_{0m,b}$ values above 8e-4 m, while some in situ sites might exhibit a locally higher $z_{0m,b}$ due to the presence of rocks or sparse vegetation elements.

Yang et al. (2008) assessed the performance of seven different parameterizations for the ratio of $z_{0h,b}/z_{0m,b}$, including Eq. 14, at several bare soil sites. Among the tested parameterizations, the formulations of Owen and Thomson (1963) and



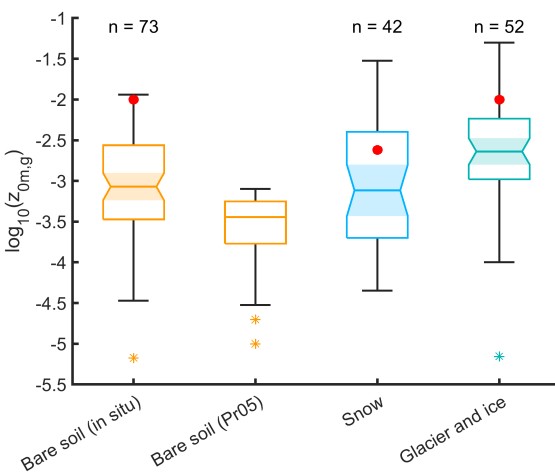

**Figure 2.** Boxplot of the decimal logarithm in in situ observations of $z_{0m,b}$ (left), $z_{0m,s}$ (second from right), and $z_{0m,i}$ (right). The value of n corresponds to the number of sites. Second from left, boxplot of $z_{0m,b}$ in remote sensing–based data of Prigent et al. (2005). Stars correspond to outliers, which are more than 1.5 times the interquartile range away from the box. Red dots show the current value in CLM5.

a revised version of Yang et al. (2002) performed best. Further, $z_{0h,b}/z_{0m,b}$ exhibits distinct diurnal variations, which is reproduced best by latter parameterization. The parameterization of Zeng and Dickinson (1998) on the other hand overestimates $z_{0h,b}/z_{0m,b}$ strongly in particular during the day. Similarly, Chen et al. (2010) implemented and tested several parameterizations

of $z_{0h,b}/z_{0m,b}$ in the Noah LSM, confirming the good performance of the formulation proposed in Yang et al. (2008) (Ya08). In particular, the Ya08 parameterization reduced the underestestimation of daytime LSTs in arid regions (Chen et al., 2011). Similar biases as for Noah exist in CLM3.5, which could be improved by decreasing $z_{0h,b}/z_{0m,b}$ (Zeng et al., 2012). Overall, there is therefore clear evidence that the parameterization of $z_{0h,b}$ and most likely also $z_{0q,b}$ applied currently in CLM5 is not ideal.

For the parameterization of $z_{0h,b}$ and $z_{0q,b}$ we therefore employ Ya08:

$$z_{0h,b} = z_{0q,b} = (70\nu/u_*) \times exp(-\beta u_*^{0.5}|T_*|^{0.25}) \qquad (15)$$

Here, $\beta$ = 7.2 and $T_*$ is the frictional temperature defined as $-SH/(\rho c_p u_*)$, where $SH$ is the sensible heat flux, $\rho$ the air density, and $c_p$ the specific heat of air at constant pressure. We have also tested the formulation of $z_{0h,b}/z_{0m,b}$ after Owen and Thomson (1963) in CLM and found no major difference to the model version using Ya08. Ya08 is also used in the revised

version of CLM to compute the $z_{0h}$ and $z_{0q}$ of snow and ice, which will be described in more detail in the next two sections.

## 2.4 Snow

The current $z_0$ representation for snow is similar to the one of bare soil. However, a globally constant $z_{0m,s}$ of 0.0024 m is used instead of 0.01 m. We here focus on $z_{0m,s}$, as the modifications of $z_{0h,s}$ and $z_{0q,s}$ were already described in the previous





section. For a comparison of $z_{0m,s}$, we collect the data compiled and measured with the wind profile method for snow in
Brock et al. (2006) as well as the measured values in Fitzpatrick et al. (2019) and van Tiggelen et al. (2021), applying the same
procedure for reported ranges as for bare soil. Again, the default value of 0.0024 m lies in the higher range of observed values,
although less drastically than for bare soil (Fig. 2). Therefore, we replace the globally constant value for $z_{0m,s}$ with the median
of 7.75e-4 m among the data from the literature.

Observations in the field show that $z_{0m,s}$ increases as melting proceeds due to the formation of melting ponds (Brock et al.,
2006; Fitzpatrick et al., 2019). Brock et al. (2006) propose the following parameterization of $z_{0m,s}$ as a function of accumulated
snow melt to account for this relation (solid line in Fig. 3):

$$ln(z_{0m,s}) = b_1 \left\{ atan\left( [log_{10}(M_a) + 0.23] / 0.08 \right) \right\} + b_4, \tag{16}$$

where $ln(z_{0m,s})$ is the natural logarithm of $z_{0m,s}$ in millimeters, $b_1$ and $b_2$ are empirical constants, and $M_a$ is the accumulated
snow melt in meters water equivalent. For application in CLM, we compute the constants $b_1$ and $b_2$ such that the parameteri-
zation will pass through the $10^{th}$ percentile of the data displayed in Fig. 2 as $M_a = 0$ m and approaches the $90^{th}$ percentile as
$M_a$ goes towards infinity, arriving at $b_1 = 1.4$ and $b_4 = -0.31$ (dashed line in Fig. 3). Additionally, $M_a$ needs to decrease again
when fresh snow falls on a snow column that was previously melting for application in a climate model. Therefore, we update
$M_a$ in CLM for snow columns that already existed at the previous time step as follows:

$$M_a^t = M_a^{t-1} - Q_{snowfall}^t + Q_{snowmelt}^t, \tag{17}$$

where $M_a^t$ and $M_a^{t-1}$ are the accumulated snow melt at the current time step and previous time step, respectively, $Q_{snowfall}^t$
is the freshly fallen snow, and $Q_{snowmelt}^t$ is the melted snow, all in meters water equivalent. Again, this parameterization of
$z_{0m,s}$ may be activated by a separate toggle, to replace to globally constant value.

## 2.5 Glaciers

The surface roughness of ice sheets and glaciers is currently the same as for bare soil. It needs to be noted that the surface
properties of land ice play a somewhat subordinate role in CLM, since they are mostly covered by snow. As for snow, we
employ the $z_{0m,i}$ observations of Brock et al. (2006), Fitzpatrick et al. (2019), and van Tiggelen et al. (2021) as a reference
(Fig. 2). The $z_{0m}$ of land ice tends to be higher than the one of bare soils or snow. Still, the current value of 0.01 m in CLM is
on the upper end of the field observations. Accordingly, we update this globally constant value to 2.3e-3 m, the median among
the collected field observations.

## 2.6 Lakes

The current lake model in CLM, the Lake, Ice, Snow, and Sediment Simulator (LISSS), was developed by Subin et al. (2012).
The $z_0$ parameterization for frozen (potentially snow–covered) lakes is consistent with ice and snow on land, as described in
the previous section. However, the $z_{0m}$ of ice was decreased in the lake model to 0.001 m, supporting the introduction of a

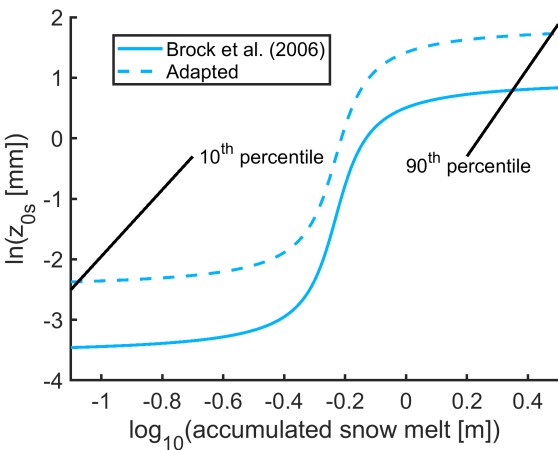

**Figure 3.** Parameterization of $z_{0m,s}$ as a function of accumulated snow melt since snow fall of Brock et al. (2006) (solid line) and parameterization with adapted constants, such that it passes through the $10^{th}$ and $90^{th}$ of data displayed in Fig. 2 (dashed line).

lower value over land, described before. For unfrozen lakes, $z_{0m}$, $z_{0h}$, and $z_{0q}$ is parameterized as follows:

$$z_{0m} = max\left(\frac{\alpha\nu}{u_*}, \frac{cu_*^2}{g}\right) \tag{18}$$

$$z_{0h} = z_{0m}exp\left(-\frac{\kappa}{P_r}\left(4\sqrt{R_0} - 3.2\right)\right) \tag{19}$$

$$z_{0q} = z_{0m}exp\left(-\frac{\kappa}{S_c}\left(4\sqrt{R_0} - 4.2\right)\right), \tag{20}$$

where $\alpha = 0.1$, $c$ is the effective Charnock coefficient (for details check Lawrence et al., 2018), $g$ the acceleration of gravity, $P_r = 0.71$ the molecular Prandtl number for air, $R_0$ the near surface atmospheric roughness Reynolds number, and $S_c = 0.66$ the molecular Schmidt number for water in air. The resultant $z_{0m}$ values over open water lie typically in the range of 1e-4 to 5e-4 m.

Subin et al. (2012) demonstrated the added value of the $z_0$ formulations described above compared to prescribing a constant value in LISSS. The WRF lake model also profited from an introduction of this parameterization (Xu et al., 2016; Wang et al., 2019). Li et al. (2015) find the dependence of $z_{0m}$, $z_{0h}$, and $z_{0q}$ on wind speed in LISSS is not ideal for a lake over the Tibetan Plateau. Still, the simulated values are generally of reasonable magnitude compared to the observed values. Further, LISSS simulated the turbulent heat fluxes at this lake still well, due to compensation of errors. Given the decent performance of LISSS also at this lake and given the fact that this study is based on measurements over one lake only, we conclude that there is no clear evidence for a need to revise the $z_0$ parameterization of LISSS. We therefore retain the current formulations for $z_0$ over lakes. We do however adopt the revisions for the $z_0$ of frozen lakes, consistent with the modifications for snow and ice on land described in Sections 2.4 and 2.5.





## 2.7 Urban areas

In the urban module of CLM, $z_0$ and $d$ are paramterized after Macdonald et al. (1998) as a function of the canyon height, $H$,
the plan area index, $\lambda_p$, and the frontal area index $\lambda_f$ (for more details see Oleson et al., 2008, 2010):

$$d = H\left(1 + \alpha^{-\lambda_p}(\lambda_p - 1)\right), \tag{21}$$

$$z_0 = H\left(1 - \frac{d}{H}\right)exp\left(-\left[0.5B\frac{C_D}{k^2}\left(1 - \frac{d}{H}\right)\lambda_f\right]^{-0.5}\right), \tag{22}$$

where $\alpha = 4.43$ is an empirical coefficient and $C_D$ is the depth–integrated mean drag coefficient for surface–mounted cubes
in a shear flow. As for vegetation, this $z_0$ corresponds to the aerodynamic $z_0$ for the exchange between the urban canopy and
the atmosphere. Again, there are additional resistance for the exchange of water vapour and energy between the surface of the
different elements in the urban environment and the urban canopy air.

Variations of $z_0/H$ among urban environments are considerable (e.g., Kanda et al., 2013). The parameterization of Macdonald et al. (1998) generally lies solidly within the spread of $z_0/H$ estimates (Grimmond and Oke, 1999; Nakayama et al., 2011;
Kanda et al., 2013). We therefore conclude that there is currently no need to revise the representation of $z_0$ and $d$ in urban
module of CLM.

## 2.8 Resulting changes in surface roughness

Here we present the alterations in $z_0$ following the mentioned model modifications in the CLM offline simulations, which
will be described in more detail in the next section. These modifications are: (1) the Ra92 parameterization with optimized
parameters based on the data of Hu20; (2) the spatially explicit input of $z_{0m,b}$ based on the data of Prigent et al. (2005); (3) the
parameterization of $z_{0m,s}$ as a function of accumulated snow melt based on the parameterization of Brock et al. (2006); (4) an
updated globally constant $z_{0m,i}$; and (5) the Ya08 parameterization for $z_{0h,g}$ and $z_{0q,g}$.

The introduction of Ra92 leads to an increase in $z_{0,v}$ for the forest PFTs (Fig. 4 a and b). In particular, the $z_{0,v}$ of forests can
increase by more than an order of magnitude during winter, because the $z_{0,v}$ of deciduous trees does no more approach $z_{0m,g}$
as they shed their leaves. Alterations of $z_{0,v}$ for grass and crops PFTs generally exhibits no clear pattern, with the exception of
a pronounced reduction in $z_{0,v}$ in the northern high–latitudes during winter (Fig. 4 c and d).

The $z_{0m.g}$ decreases by more than an order of magnitude in most cases due to our revisions of $z_{0m,b}$, $z_{0m,s}$, and $z_{0m,i}$ (Fig. 5 a
and d). Only in some coastal areas of Greenland $z_{0m,g}$ increases slightly, as enough snow melt accumulates to reach the higher
end of the Brock et al. (2006) parameterization for $z_{0m,s}$. The $z_0$ for scalars ($z_{0h,g}$ and $z_{0q,g}$) now exhibit a distinct diurnal
cycle following the introduction of Ya08. It increases at daytime in low–latitudes and during summer in the mid–latitudes, but
decreases under stable conditions often present in high–latitudes and at night.

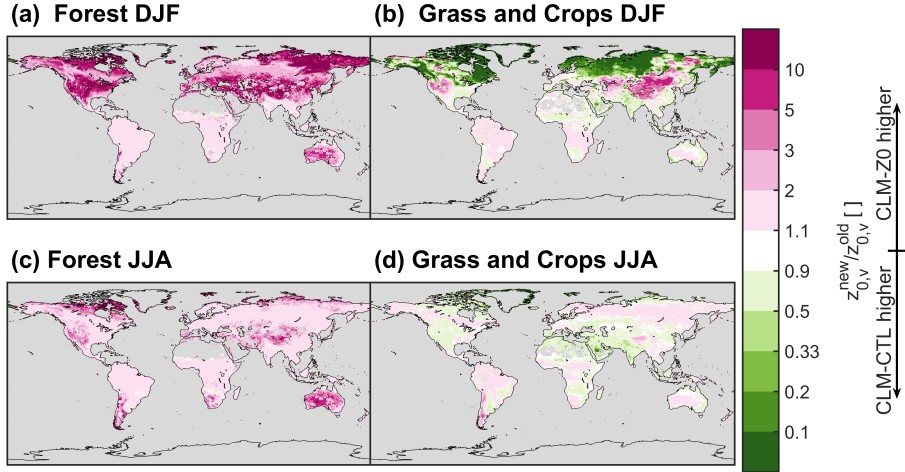

**Figure 4.** Ratio of new vegetation surface roughness ($z_{0,v}$; in CLM–Z0) divided by old $z_{0,v}$ (in CLM–CTL). Panels (a), (c) ratio of average $z_{0,v}$ across forest plant functional types and (b), (d), across grass and crop plant functional types. Upper row boreal winter (DJF) and lower row boreal summer (JJA).

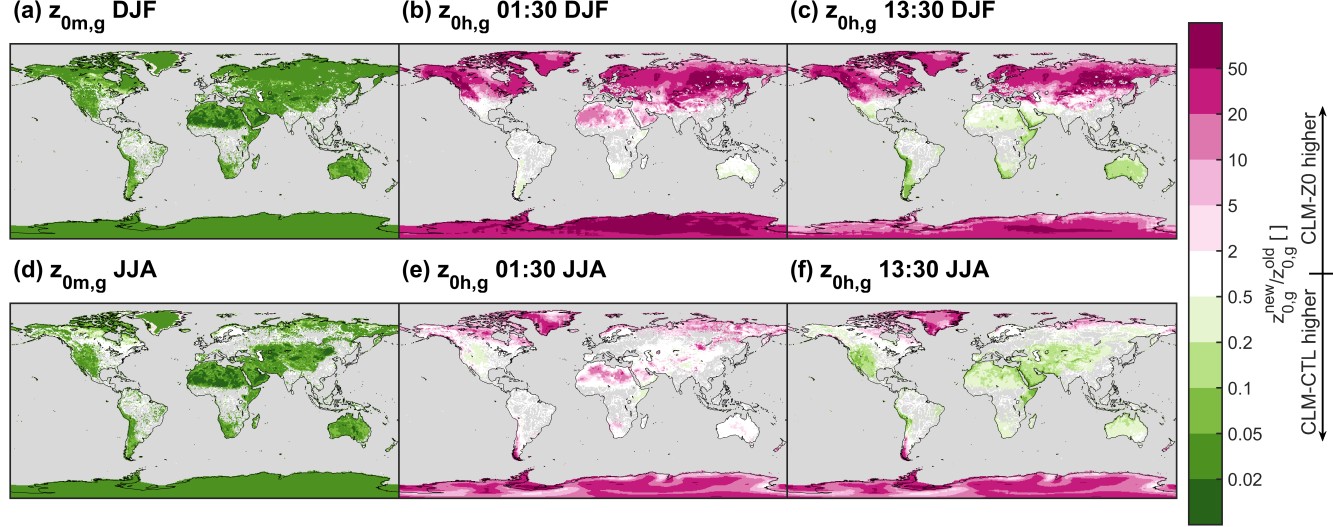

**Figure 5.** Ratio of new ground surface roughness ($z_{0,g}$) divided by old $z_{0,g}$. Panels (a), (d) momentum surface roughness, (b), (e), surface roughness of scalars at 01:30 local solar time, and (c), (f), surface roughness of scalars at 13:30 local solar time. Upper row boreal winter (DJF) and lower row boreal summer (JJA).





## 3 Experiment design

In this study, we present results from two sets of simulations: (1) Land–only (offline) simulations using CLM version 5.1 forced by the GSWP3 reanalysis data (Dirmeyer et al., 2006; Kim, 2014) and (2) land–atmosphere (coupled) simulations with
CESM version 2.1.2. For each simulation, we conduct a 50–year spinup followed by a 15–year analysis period using a near present–day climatological configuration. The vegetation phenology is prescribed from satellite observations in all simulations (Sp–mode). The different patches of vegetation are placed on separated soil columns to suppress lateral exchange of energy and water among them (Schultz et al., 2017; Meier et al., 2018) and biomass heat storage was activated to remove the stability cap of the Monin–Obukhov stability parameter (Swenson et al., 2019; Meier et al., 2019). Besides, we implement a new history
file averaging flag, which interpolates linearly in time to retrieve model output at the specified local solar time. This allows to determine the model state for example always at 01:30 without outputting the variables of interest at all model time steps, avoiding both excessive use of storage space and a cumbersome post–processing of the data. The model output at a specific local solar times allows to examine diurnal variations of various variables and is further used for comparison to the MODIS LST observations, which are made at approximately 01:30 and 13:30 local solar time. For each set up we conduct one control
simulation with the current representation of $z_0$ in CLM and a simulation in which the updates for $z_0$ as described in the previous section were activated.

For the CLM simulations, we use the component configuration set "I2000Clm51Sp". These simulations are run at 0.5° resolution. For the atmospheric forcing we cycle through the GSWP3 data of 1998–2012. The resulting simulations are called CLM–CTL and CLM–Z0 subsequently. In addition, a series of CLM experiments is presented in Appendix A1 to assess the
effect of the individual modifications. Table A1 provides an overview of all CLM simulations.

The CESM simulations are run in the configuration "F2000climo" at 0.9°x1.25° resolution. This configuration couples CLM version 5.0 with the atmospheric model CAM version 6.0. The ocean is prescribed in F2000climo from HadISST v1.1 (i.e., it is run in data mode; Hurrell et al., 2008). For the prescribed sea surface temperature forcing we cycle through the data of 1998–2012 instead of using the data from 2000 only, as normally the case in F2000climo. This is done to introduce more
interannual variability. We call the CESM simulations CESM–CTL and CESM–Z0 subsequently.

## 4 Model analysis and evaluation

### 4.1 Reference data sets

We consult two observation–based data sets to assess the impact of the imposed modifications in CLM–Z0 and CESM–Z0 on model performance in terms of the land surface temperature (LST). First, we use observations of the MODerate resolution
Imaging Spectroradiometer (MODIS) system, which is installed on the low–earth orbit satellites Terra and Aqua to evaluate diurnal variations of the LST at grid cell level. These instruments provide LST estimates at a resolution of 1 km at approximately 01:30 and 13:30 local solar time at the equator, based on the longwave radiation emitted by the land surface. We employ data from 2002–2012 of the product MYD11C3 version 6 (Wan et al., 2015). which has a native resolution of 0.05° degree. From





this data we compute a multi–year monthly climatology as described in Meier et al. (2019) at 0.5° resolution. For comparison

to the CESM simulations, we regrid this climatology to 0.9°x1.25° with first–order conservative remapping of the Climate Data Operators library (CDO). We output the LST in the model simulations at 01:30/13:30 and use only model output for 2002–2012 for a consistent comparison with MODIS. Further, we apply a cloud masking to the model output as described below.

In addition to comparing LST directly at grid cell level, we also evaluate the local LST difference between bare soil and

vegetation. To extract such information from the MODIS observations, we repeat the space–for–time substitution approach as in Duveiller et al. (2018) for the conversion of all types of vegetation to bare soil. We conduct a multiple linear regression between MODIS LST observations and grid–level land cover fractions within a moving window of 5 by 5 pixels for each month in 2008–2012. For the LST, we employ monthly MYD11C3 data both at daytime (1̃3:30 local solar time at the Equator) and nighttime (0̃1:30 local solar time at the Equator). The land cover fractions are based on the ESA Climate Change Initiative

Land Cover project (ESA, 2017). To estimate the potential change in LST for a conversion between vegetation and bare land, we aggregate all land cover types that involve vegetation to one land cover class and focus on the slope of the multiple linear regression between the resultant vegetated land cover class and bare land. With this procedure we retrieve a monthly observation–based estimate of the LST sensitivity to a conversion of vegetation to bare land at 0.25° resolution, along with an estimation of the retrieval uncertainty associated with the regression (see Duveiller et al. (2018) and Duveiller et al. (2021) for

details). For comparison to the CLM simulations, we compute the multi–year monthly average at 0.5° resolution, weighing all grid cells that fall into the focal location–month combination by area and by 1 over the uncertainty estimate of the respective value. In CLM, we compute the sub–grid difference in the variable of interest of the bare soil patch minus all vegetation patches (including crops) within a grid cell as described in more detail in Meier et al. (2018). Again we only use cloud–masked data for 2008–2012 LST, which was output at 01:30 and 13:30 local solar time.

**4.2 Cloud masking**

MODIS can observe the LST only under clear–sky conditions (Wan et al., 2015). We therefore remove cloudy conditions in our model output when confronting it with MODIS. For the CESM simulations, we can filter for clear–sky conditions directly from the total cloud cover model output. To do so, we output the total cloud coverage and the variables of interest at daily temporal resolution. In the post–processing we then remove days with an average total cloud coverage above 50 %. It is more

complex to exclude cloudy days in the offline CLM simulations, since the GSWP3 forcing does not include information on cloud coverage (Kim, 2014). We therefore mask for cloudy days based on the incoming shortwave radiation. This is done through a comparison to the theoretical daily incoming solar radiation at the top of the atmosphere according to Berger (1978), $W_{TOA}$, which is a function of latitude and the day of the year. However, solar radiation passing through the atmosphere can be altered even under clear–sky conditions for example because of aerosols (IPCC, 2013). Therefore, we derive a climatology of

the incoming solar radiation at the surface, $W_S^{cs}$, based on $W_{TOA}$ in an iterative procedure:





1. A multiplicative factor, $C$, is optimized, such that it minimizes the sum–squared deviation to the daily incoming solar radiation forcing of GSWP3 at a given location:

$$W_S^{cs} = C \cdot W_{TOA} \tag{23}$$

2. Incoming solar radiation values below $80\,\%$ of $W_S^{cs}$ are removed for the next iteration, unless the current fit is based on less than 200 values (the iteration starts with $15 \cdot 365 = 5475$ values).

3. This iteration is stopped if the sum–squared deviation of $W_S^{cs}$ to the remaining daily incoming solar radiation forcing of GSWP3 improves by less than $10\,\mathrm{W^2 m^{-4}}$.

With this procedure we estimate $W_S^{cs}$ for each land point. We then remove days where the daily incoming solar radiation lies below $20\ \mathrm{Wm^{-2}}$ or $90\,\%$ of $W_S^{cs}$ in the post–processing of the model output of the CLM simulations. Fig. 6 illustrates this clear–sky masking for four grid cells. Note that this cloud–masking procedure is not perfect because it effectively ignores clouds at night and does not distinguish between cloud types, which affect the incoming shortwave radiation at the surface differently (L'Ecuyer et al., 2019). Also, it results in data gaps in the masked data during the polar night, because no incoming shortwave radiation is available for the cloud masking procedure.

### 4.3 Significance testing

The CESM simulations exhibit a considerable interannual variability. Therefore, we conduct a statistical test to assess whether the identified seasonal differences between CESM–Z0 and CESM–CTL are significant. For the sample of 14 seasonal mean differences between CESM–Z0 and CESM–CTL for each grid cell and season we make a one–sample student's t–test at $5\,\%$ confidence level. This test in isolation is inappropriate when applied to a spatially auto–correlated field, as clustered areas can appear erroneously significant (Wilks, 2016). Thus, we control the false discovery rate as proposed in Wilks (2016) using a confidence level of $10\,\% (= 2 \cdot 5\,\%)$, which is appropriate for data with a moderate to strong spatial auto–correlation. In addition, we include the last 30 years of the spinup period for some variables to corroborate the presented results.

## 5   Results

We first focus on the LST response at 01:30/13:30 local solar time in the land–only CLM simulations in Section 5.1. In this section, we also evaluate the simulated diurnal variations in LST compared to MODIS and the LST sensitivity to a conversion of vegetation to bare land compared to Du18. In Section 5.2 we assess the response to the imposed $z_0$ modifications in the CESM land–atmosphere simulations. Initially, the focus is again on the LST (Section 5.2.1) and additionally the air temperature at the bottom of the atmosphere (Section 5.2.2). Afterwards, we present alterations in wind speed. Note that we present a number of sensitivity experiments in Appendix A1, where we assess the influence of the different $z_0$ modifications individually. Further, we conduct an energy balance decomposition after Luyssaert et al. (2014) in Appendix A2 to link the changes in LST described in this section to individual energy fluxes.



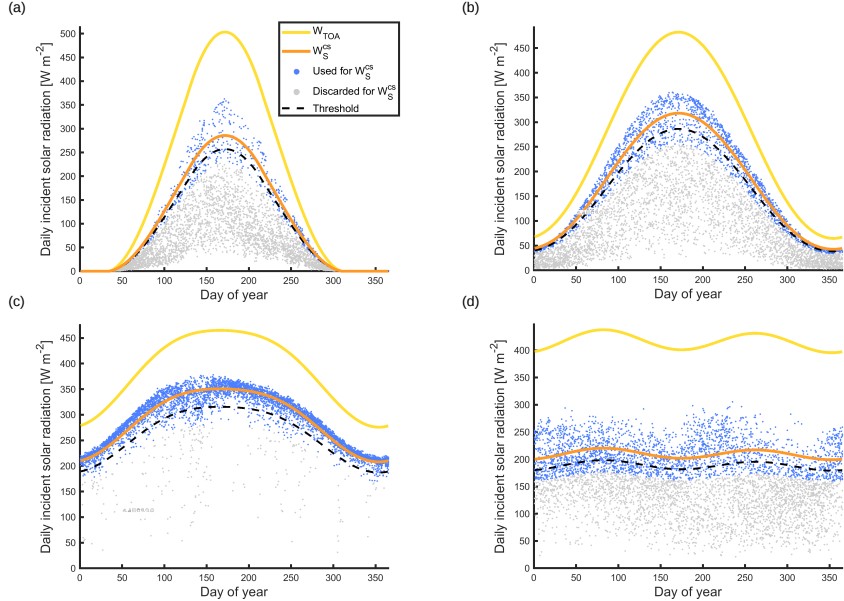

**Figure 6.** Examples of cloud masking based in incoming shortwave radiation at (a) $73.25°$ N/$11.75°$ E, (b) $53.25°$ N/$11.75°$ E, (c) $23.25°$ N/$11.75°$ E, and (d) $3.25°$ N/$11.75°$ E. Yellow line daily incoming solar radiation at the top of atmosphere according to Berger (1978), orange line fitted incoming shortwave radiation at surface under clear–sky conditions, blue dots daily incoming solar radiation values in GSWP3 included to make this fit, grey points daily incoming solar radiation values in GSWP3 removed because they are below $80\,\%$ of the last fit of $W_S^{cs}$, and dashed black line threshold of $90\,\%$ of $W_S^{cs}$ above which days are considered clear–sky.

## 5.1 LST response in land–only simulations

At 13:30 the LST increases substantially in warm desert regions (Fig. 7 a and c). This warming originates mainly from the reduction in $z_{0m,g}$, while the introduction of the Ya08 formulation for $z_{0h,g}$ and $z_{0q,g}$ produces only a small impact (Appendix A1). The reduced $z_{0m,g}$ inhibits the exchange of sensible heat with the atmosphere (Fig. A2). The solar radiation absorbed by the land surface in desert regions is therefore transferred less efficiently to the atmosphere in CLM–Z0 than in CLM–CTL. Consequently, the land surface warms and maintains its energy balance through emission of more longwave radiation and a higher ground heat flux (Fig. A2). Accordingly, the induced warming is higher during the summer season, when the solar irradiance is highest. On the other hand, the reduction in $z_{0m,g}$ decreases the LST in the cold deserts, in particular during the winter season. This is again the result of a reduced sensible heat flux, which is however generally directed from the warmer atmosphere to the land surface in those regions. In vegetated areas, the increased $z_{0,v}$ of forests enhances the turbulent transport of energy away from the land surface (Fig. A2), producing a cooling of the daytime LST.

The LST response at 01:30 is generally considerably weaker than the daytime effect (Fig. 7 b and d). Conditions in the surface layer are more commonly stable at night than at day, which inhibits the turbulent energy exchange between the land





and the atmosphere. Therefore, our modifications of $z_0$ produce a weaker effect. Interestingly, the pronounced daytime warming

effect in the warm deserts translates into the night through the energy stored in the soils (Fig. A3). In contrast, the increase in

$z_{0,v}$ of forests warms the land surface at night in particular during summer by increasing the sensible heat flux towards the

land. Thus, the LST response at 01:30 over vegetation opposes the daytime response in sign, unlike in desert regions. This is

likely the case, because the LST in CLM is linked tightly to the vegetation temperature (Meier et al., 2019), which exhibits a

smaller thermal inertia than the ground. Consequently, the alterations in LST change sign diurnally in regions dominated by

vegetation, while the sign remains the same over regions dominated by bare soils.

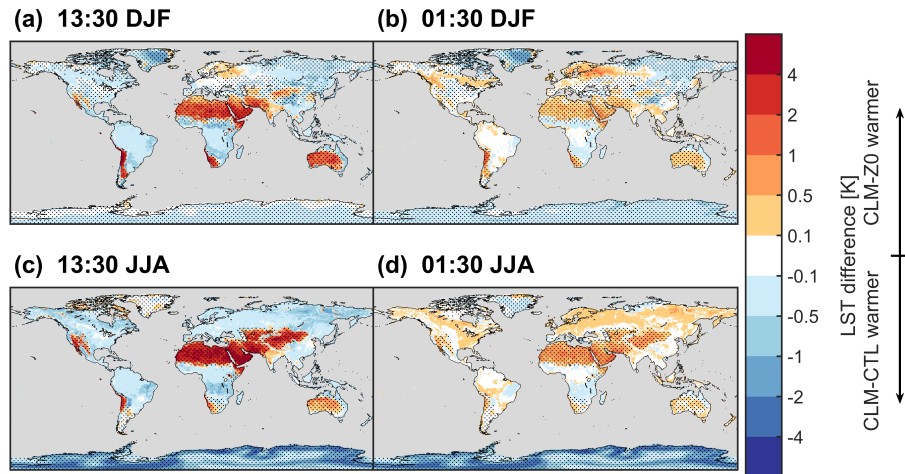

**Figure 7. LST difference between CLM–Z0 and CLM–CTL.** Left column LST difference at 13:30 local solar time and right column difference at 01:30 local solar time. Upper row boreal winter (DJF) and lower row boreal summer (JJA). The stippling shows areas dominated by bare soil with a seasonal average $VAI$ below $0.5\,\mathrm{m^2\,m^{-2}}$. Note the non–linear colour scale.


Overall, the modified $z_0$ amplify the diurnal temperature range (DTR, here defined as the LST difference between 13:30 and

01:30 local solar time) in desert regions and dampen the DTR in regions with forests (Fig. 8 a). This links back to previous

studies that found an overestimation of the DTR in desert regions and an underestimation over forests in CLM compared to

remote sensing observations (Zeng et al., 2012; Meier et al., 2019). This tendency prevails in the current version 5.1 of CLM

(Fig. 8 d). The modifications of $z_0$ in CLM–Z0 alleviate the mentioned biases in most regions with the notable exception of the

southern half of the Sahara, where the reduced $z_{0m,g}$ in CLM–Z0 frequently overcompensates an only slight underestimation

of the LST DTR in CLM–CTL (Fig. 8 b, c, e, and f).

The modifications in CLM–Z0 also affect the sensitivity of the LST to land cover. Here we compare the LST sensitivity for

converting vegetated land to bare soil as estimated in Du18 to the subgrid LST difference between the bare soil tile and the

vegetated tiles in CLM. This land cover transition could be relevant for the biogeophysical response to desertification, which

has become more common over the last decades (IPCC 2019). Overall, Du18 observes an increase in LST at 13:30 over bare

soils compared to vegetation with the exception of latitudes exceeding 40° N/S during the colder months (Fig. 9 a). CLM–CTL



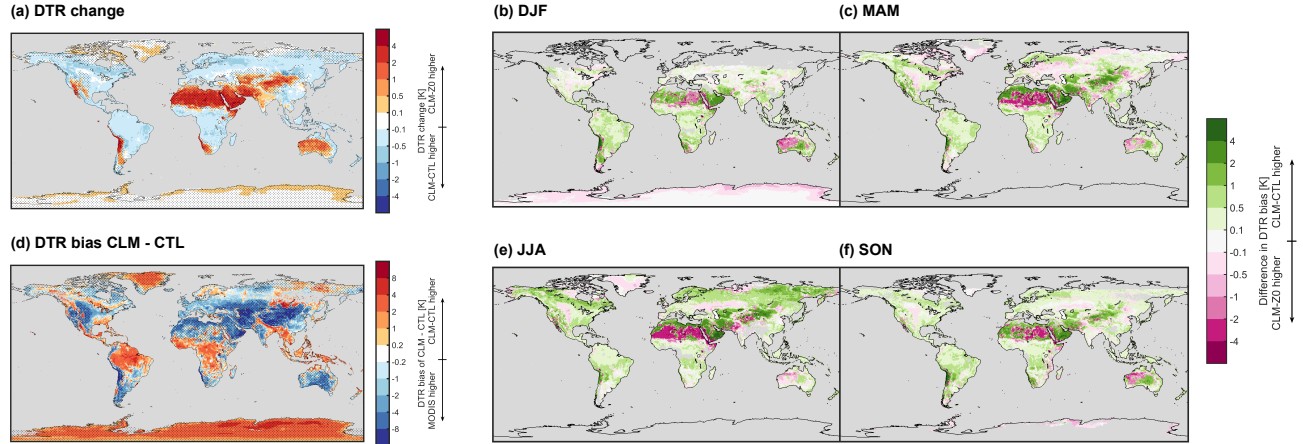

**Figure 8.** Panel (a), difference in LST diurnal temperature range (DTR) of CLM–Z0 minus CLM–CTL and panel (d) bias in LST DTR of CLM–CTL compared to MODIS remote sensing observations. The stippling in those panels shows areas with an average $VAI$ below $0.5\,\mathrm{m^2\,m^{-2}}$. To the right, change in the LST DTR bias between CLM–Z0 and CLM–CTL in boreal winter (b), spring (c), summer (e), and autumn (d). CLM data are cloud–masked based on the incoming shortwave radiation. Note the non–linear colour scale.

on the other hand exhibits a lower daytime LST over the bare soil tiles than over the vegetated tiles in most cases (Fig. 9 b). CLM–Z0 captures the LST increase at 13:30 in most cases (Fig. 9 c). However, the signal in the latter simulation is considerably
stronger than in Du18, resulting in a higher RMSE for this simulation than in CLM–CTL. At night, the modifications in CLM–Z0 further amplify a positive bias in the LST difference between bare land minus vegetation of CLM–CTL in comparison to Du18 (Fig. 9 e–h). For the DTR, Du18 finds an amplification over bare land compared to vegetation for most latitude–month combinations, with the exception of the high–latitudes during winter Fig. 9 j). CLM–CTL on the other hand mostly exhibits a lower DTR over bare soils than over vegetation (Fig. 9 k). This bias is mitigated to some extent in CLM–Z0 even
though a dampening of the DTR often persists in the northern mid–latitudes (Fig. 9 l). Overall, the imposed alterations in $z_0$ do not result in a clear improvement of the LST sensitivity to a conversion between vegetation and bare soil in CLM, but clearly alter this sensitivity. Note that some discrepancies between Du18 and the CLM simulations might also arise from the neglect of atmospheric feedbacks due to the sub–grid approach in CLM (note that the sub–grid approach would still neglected atmospheric feedbacks in the CESM simulations; for more information see Chen and Dirmeyer, 2020). In addition, the cloud–
masking based on the incoming solar radiation could potentially introduce errors in CLM, in particular for the nighttime signal. Further, preferential occurrence of clouds over vegetation or bare soil might introduce biases in Du18. In fact, a recent study observed increased low level cloud cover over forests compared to short vegetation, using a similar methodology as in Du18 (Duveiller et al., 2021).











**Figure 9.** (Previous page) **LST sensitivity in Du18 and CLM to conversion of vegetation to bare land.** Panels (a)–(d), LST difference between bare soil minus vegetated land at 13:30 local solar time ($\Delta\mathrm{LST}_{max}$). Seasonal and latitudinal variations of ($\Delta\mathrm{LST}_{max}$) in (a) the observation–based estimate of Du18, (b) CLM–CTL, and (c) CLM–Z0. Points with a mean which is significantly different from zero in a two–sided t–test at 95% confidence level are marked with a black dot. All data from the 2008–2012 analysis period corresponding to a given latitude and a given month are pooled to derive the sample set for the test. The numbers next to the titles are the area–weighted spatiotemporal root–mean–squared deviation of the respective simulation against Du18. Panel (d) shows the zonal annual mean of Du18 (black, range between the $10^{th}$ and $90^{th}$ percentiles in gray), CLM–CTL (blue, range between the $10^{th}$ and $90^{th}$ percentiles in blue), and CLM–Z0 (red, range between the $10^{th}$ and $90^{th}$ percentiles in orange). Note that on this subfigure results have been smoothed latitudinally with a simple moving average over $4°$. CLM data are cloud–masked based on the incoming shortwave radiation. Panels (e)–(h) the same for the LST difference at 01:30 local solar time and panels (j)–(m) for the diurnal temperature range.

## 5.2 Effect in land–atmosphere coupled simulations

So far, we have assessed the effect of the alterations in $z_0$ in CLM simulations forced by the GSWP3 reanalysis data. However, the resultant alterations of the turbulent fluxes at the land surface may also affect the atmosphere, which is neglected in land–only simulations. Therefore, we present the effect of the imposed $z_0$ modifications in land–atmosphere coupled simulations using CESM in this section.

### 5.2.1 LST response

At low latitudes, the LST at 13:30 in CESM–Z0 increases over the deserts and decreases in most regions with dense vegetation similar to the offline simulations (Fig. 10 a and b). However, the daytime warming in deserts is stronger in CESM than in CLM (Fig. 7). It therefore appears that atmospheric feedbacks trigger an additional warming of the land surface in these regions. Indeed, we find an increase in incoming shortwave radiation accompanied by a reduction in cloud cover most notable over the Sahara and the Middle East (Fig. 10 e–h and Figs. A4 and A5). An increase in cloud coverage as a consequence of an
increase in the sensible heat flux was found in previous studies (Khanna et al., 2017; Bosman et al., 2019). It is therefore possible that the reduction in cloud coverage over desert regions in CESM–Z0 is a by–product of the lower sensible heat flux in this simulation. Over the northern mid- and high–latitudes, a reduction in cloud cover during summer coincides in turn with reduction in daytime LSTs in CESM–Z0 due to less incoming shortwave radiation (Figs. A4). The LST response at night is often weaker but of the same sign as the daytime signal in CESM, similar to the offline simulations (Fig. 10 c and d). However,
no distinct nighttime warming emerges over mid–latitude forests during the summer season at night in CESM, which was the case in CLM (compare Figs. 7 d and 10 d). In the mid- and high–latitudes, changes in LST often exhibit a similar spatial pattern to surface air temperature changes, which are discussed in more detail in the next section (compare Fig. 10 and Fig. A6). In particular, the warming of the LST during winter in CESM–Z0 appears to be related to more incoming longwave radiation at the land surface (Fig. A5).

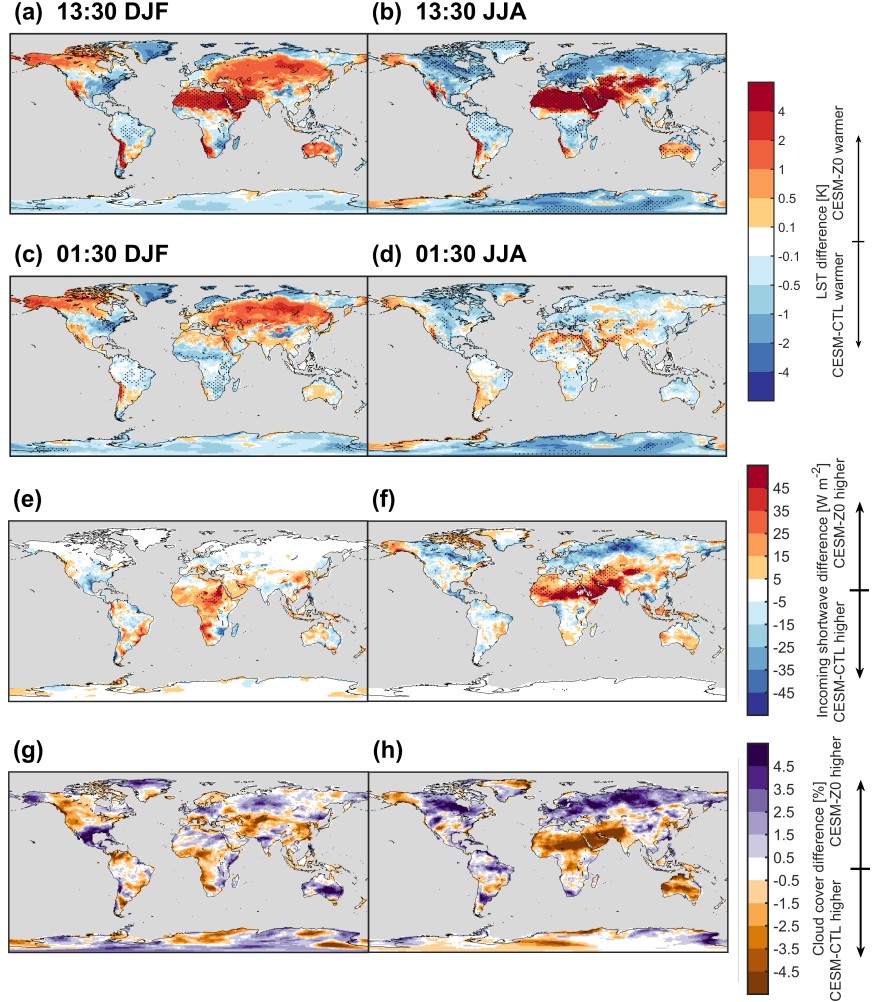

**Figure 10.** LST difference between CESM–Z0 and CESM–CTL at (a), (b) 13:30 local solar time and (c), (d) 01:30 local solar time. Panels (e) and (f), difference in incoming shortwave radiation at 13:30 local solar time between CESM–Z0 and CESM–CTL and bottom row difference in daily average total cloud cover. The stippling shows areas with a difference that is statistically significant different from zero in a two–sided t–test at 95% confidence level with a controlled false discovery rate. Left column boreal winter (DJF) and right column boreal summer (JJA). Note the non–linear colour scale for panels (a)–(d).

Compared to the MODIS observations, CESM–CTL underestimates the DTR in LST in most areas with the notable exceptions of the polar regions and parts of the Amazon (Fig. 11 b). As the case in the offline simulations, this underestimation is most distinct in the warm deserts. Again, the reduced $z_{0m,g}$ amplifies the DTR in those desert regions producing an improved agreement with the remote sensing observations (Fig. 11). Apart from these regions, the results are more mixed. Still, there is a clear improvement over the northern mid–latitudes during boreal summer. Yet, the alterations of $z_0$ in CESM–Z0 alone do





not alleviate the widespread underestimation in the LST DTR of CESM entirely (Fig. A7). The remaining biases may not only
      originate from deficiencies at the land surface itself but could also be related to atmospheric components such as the radiation
      scheme.

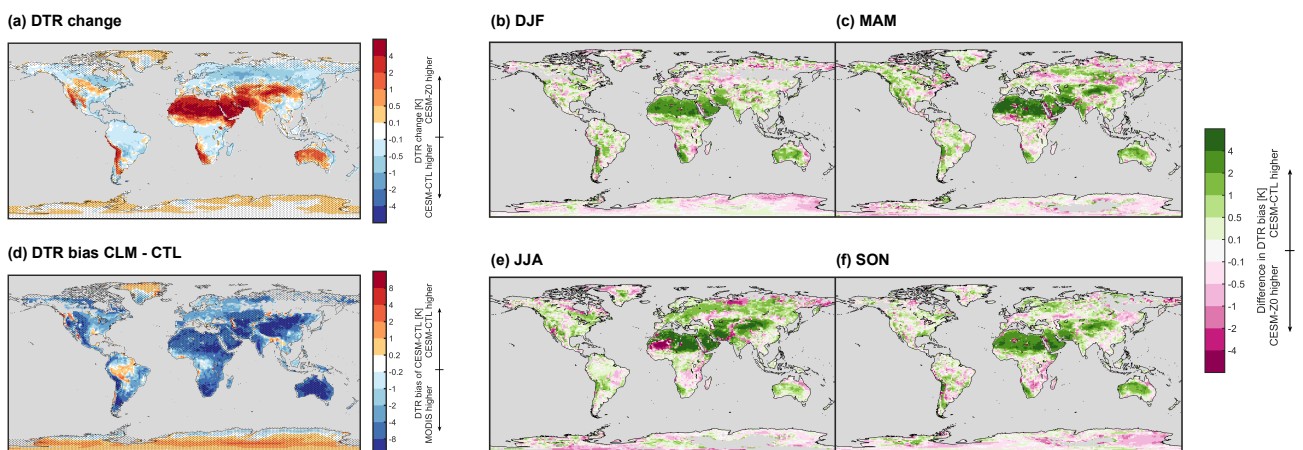

**Figure 11.** As Fig. 8 but for land–atmosphere coupled simulations CESM–Z0 and CESM–CTL. CESM data are cloud–masked.

### 5.2.2   Response in surface air temperature and comparison to LST

The altered surface energy fluxes thus also affect air temperatures at the bottom of the atmospheric column (TBOT). The
difference in daily average TBOT between CESM–Z0 and CESM–CTL exhibits considerable interannual variability. Therefore,
      we included the last 30 years of the spinup period to corroborate the results shown in Fig. 12 (a) and (b). Fig. A6 depicts the
      average TBOT response for the analysis period and the last thirty years of the spinup period separately. Even when including
      these additional years some pronounced features, such as the wintertime warming of average TBOT over North Asia, are still
      not statistically significant. Nevertheless, the wintertime average TBOT increases considerably in many regions in the northern
hemisphere, showing a similar spatial pattern as the LST response (Fig. 12 a). This is linked to more incoming longwave
      radiation (Fig. A5). On the other hand, the increase in $z_{0,v}$ decreases the summertime TBOT in those regions (Fig. 12 b). This
      can be explained by lower incoming shortwave radiation in CESM–Z0 compared to CESM–CTL (Fig. A4) as a result of higher
      total cloud coverage (Fig. 10 e). Consequently, less energy is available close to the land surface in CESM–Z0, cooling both the
      LST and TBOT. At low–latitudes, TBOT decreases mostly over the rain forests. Interestingly, CESM–Z0 also often exhibits
a lower average TBOT over the Sahara in particular during boreal winter, thus opposing the LST response in sign. Further,
      there is a distinct band where TBOT warms in JJA over the Sahel region, while it cools both just north and south of this region,
      which emerges both during the analysis period and during the last 30 years of the spinup (Fig. A6).

      The effect on the DTR of TBOT in CESM–Z0 opposes the effect on the LST DTR in sign, which is best visible in Africa
      (compare Fig. 12 c and d to Fig. 11 a). In case of a decrease in $z_0$, less energy is transferred from the land surface into the





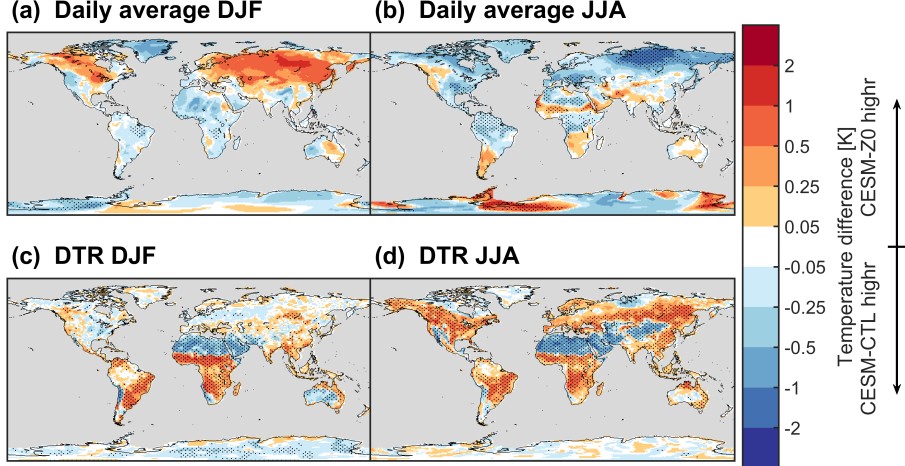

**Figure 12.** Panels (a) and (b), seasonal average difference in air temperature at the bottom of the atmospheric column (TBOT) between CESM–Z0 and CESM–CTL using data from the last 30 years of the spinup period and data from the analysis period (15 years). Below difference in TBOT diurnal temperature range (DTR). The stippling shows areas with a difference that is statistically significant different from zero in a two–sided t–test at 95% confidence level with a controled false discovery rate. Left column boreal winter (DJF) and right column boreal summer (JJA). Note the non–linear colour scale.

atmosphere under unstable surface layer conditions (which are frequently present during day) and from the atmosphere to the land surface under stable conditions (frequently present at night). Consequently, the DTR at the land surface (LST) is amplified, while the DTR is dampened in the atmosphere above. This dipole between the DTR response of LST and TBOT to alterations in $z_0$ was previously found also in the context of deforestation in CESM (Chen and Dirmeyer, 2019) and in a number of regional climate models (Breil et al., 2020).

Fig. 13, displays how the response of the DTR in LST and TBOT scale with the change in $z_{0m}$. The DTR in LST for the individual vegetation patches (PFTs) decreases linearly with the logarithm of the ratio between the $z_{0,v}$ in CESM–Z0 and the $z_{0,v}$ in CESM–CTL, with a slope of -3.1 K (when using the decimal logarithm; Fig. 13 a). In other words, a tenfold increase in $z_{0,v}$ dampens the DTR by 3.1 K. At grid cell level, the LST DTR reacts comparably strong to the relatively small changes in $z_{0m}$ by a factor of 3 or less, as visible by values between -0.5 to 0.5 on the x–axis in Fig. 13 b. For stronger reductions in $z_{0m}$

over desert regions the amplification of the LST DTR saturates at approximately 4 K. This scale dependence likely originates from several factors. First, smaller changes in $z_{0m}$ in CESM–Z0 compared to CESM–CTL occur over vegetation, while the strong reductions occur over bare soil (compare Fig. 4 to Fig. 5 a and d). It might therefore be that the LST reacts stronger to alterations of $z_{0,v}$ than to alterations of $z_{0m,g}$ due to the smaller thermal inertia of vegetation compared to soils. Second, different types of land cover with varying changes in $z_{0m}$ are mixed at grid cell level. For some PFT patches, $z_{0,v}$ increases by

more than an order of magnitude (i.e., $log_{10}(z_{0,v}^{new}/z_{0,v}^{old}) > 1$), which is never the case for entire grid cells. Third, our sensitivity experiments in Appendix A2 show that the concurrent reduction of $z_{0m,g}$ with the alterations $z_{0,v}$ amplify the response of the



LST DTR over vegetation, compared to a simulation were only $z_{0,v}$ changed. And forth, the sensitivity experiments indicate that the introduction of Ya08 for $z_{0h,g}$ and $z_{0q,g}$ moderates the LST DTR response to the decrease in $z_{0m,g}$ over the Sahara. Again, the dipole between the LST DTR response and the TBOT DTR response can be observed when comparing panels (b)

and (c) in Fig. 13. The two variables are clearly mirrored in sign. However, the response in TBOT DTR is considerably weaker than the one of LST. This is likely owed to the differing nature of these two variables. The LST is computed from longwave radiation emitted by the land surface and is therefore tightly coupled to the energy redistribution at the land surface. TBOT is in contrast affected not only by the energy redistribution at the land surface, but also by lateral and vertical mixing of air masses. This mixing may explain why the TBOT DTR response is generally weaker than the LST DTR response.

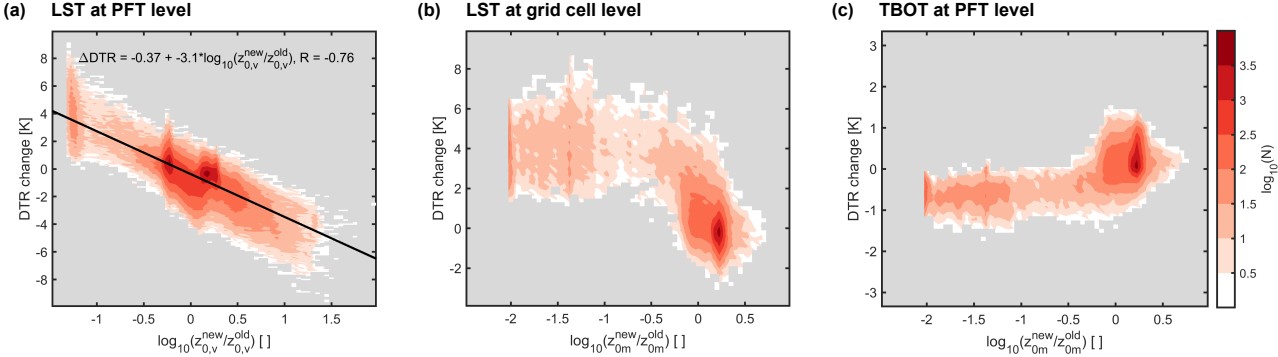

**Figure 13.** Panel (a), density plot of change in multi–year monthly mean LST DTR at PFT–level of CESM–Z0 minus CESM–CTL versus the decimal logarithm of the ratio of $z_{0,v}$ in CESM–Z0 divided by $z_{0,v}$ in CESM–CTL. Binsize on x–axis is 0.05 and on y–axis 0.1 K. Colour scale on the very right shows the decimal logarithm of the number of tiles that fall within the respective bin. Multi–year monthly mean data of all PFTs excluding bare soil between 30° N/S was used to generate this figure. Panels (b) and (c), the same for the LST DTR (b) and TBOT DTR (c) at grid cell–level and the maximum of $z_{0m,g}$ and $z_{0,v}$. Bin size on y–axis in panel (c) is 0.05 K. Black line in panel (a) shows linear fit with its formula and the Pearson correlation coefficient (R) above. Note the differing ranges of the y–axis for the different panels.





### 5.2.3 Response in surface wind speed

So far, our analysis was focused on temperatures at and above the land surface. The identified temperature changes in CLM–Z0 and CESM–Z0 are closely linked to alterations of the surface energy redistribution, even though some contributions from atmospheric feedbacks emerged in the coupled simulations. However, the modifications in $z_{0m}$ also affect the drag exerted by the land surface and thereby most likely wind speeds, at least close to the surface.

Indeed the wind speed at the lowest atmospheric level increases notably in CLM–Z0 over desert regions, where $z_{0m}$ was lowered (Fig. 14 a and e). The remaining land mass is dominated by reductions in surface wind speed, consistent with the increase in $z_{0,v}$ introduced for most vegetation types in CLM–Z0. These alterations of surface wind speed decay relatively fast with height and are only rarely significant at a height of 1.1 km (Fig. 14 b and f). Even over the Sahara, where wind speeds close to the surface increase considerably, this signal disappears about 2.5 km above the surface (Fig. 14 d). There are also few regions over the oceans where CLM–Z0 exhibits significant changes in surface wind speed. Unlike wind speed changes over land, these features are present even stronger at higher altitudes (Fig. 14 g and h). This makes sense as the $z_{0m}$ over oceans was not modified in CESM–Z0. Therefore, surface wind speed alterations over oceans are driven by wind speed changes higher up rather than alterations of the surface (momentum) fluxes.

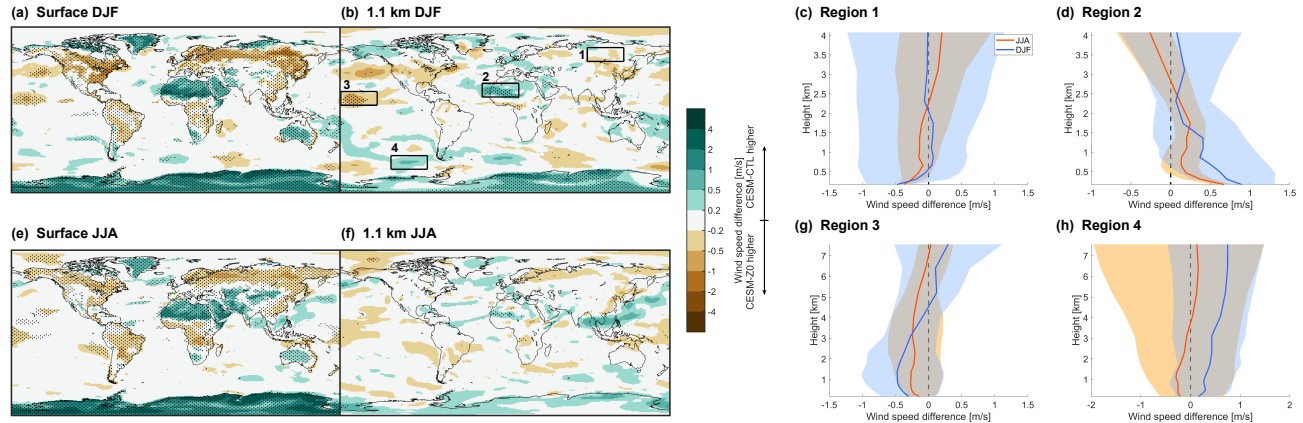

**Figure 14.** Seasonal mean wind speed difference of CESM–Z0 minus CESM–CTL at lowest atmospheric level (a, e) and approximately 1.1 km above sea-level (b, f). Top row, boreal winter (DJF) and bottom row boreal summer (JJA). The stippling shows areas with a difference that is statistically significant different from zero in a two–sided t–test at 95% confidence level with a controlled false discovery rate. Note the non–linear colour scale. Panels (c), (d), (g), and (h), profile of area–weighted mean wind speed difference in DJF (blue) and JJA (red) in regions 1 (c), 2 (d), 3 (g), and 4 (h), which are marked in panel (b). Line depicts median wind speed difference across all seasonal means and shading range between $10^{th}$ and $90^{th}$ percentile. Height is calculated assuming a surface pressure of 1013.2 hPa, a surface air temeprature of 288.15 K, and a constant lapse rate of 6.5 K km$^{-1}$. Data from the last 30 years of the spinup period and data from the analysis period (15 years) were used for this figure.


## 6   Conclusions

In this study, we have compared the representation of $z_0$ in CLM to observations and parameterizations that exist in the literature, conducted revisions of CLM when clearly supported by this comparison, and assessed the impact of these revisions on simulated temperatures at the land surface and wind speed. Specifically, we introduced the parameterization proposed by Raupach (1992) for the $z_0$ of vegetation, where parameter choices were optimized such that the parameterization conforms with the observational data of Hu et al. (2020). The $z_0$ of forests is increased considerably with this new parameterization, while the

one of crops is decreased. Further, the $z_0$ of broadleaf deciduous forests exhibits now a minimum during the growing phase as observed in several studies. The globally constant value for $z_{0m}$ over bare soil, snow, and glaciers of the default version of CLM is clearly overestimated in comparison the observations collected from the literature. Therefore, $z_{0m}$ is decreased from 1e-2 to 8.4e-4 m, from 2.4e-3 to 7.8e-4 m, and from 1e-2 to 2.3e-3 m for bare soil, snow, and glaciers, respectively. Alternatively, the spatially explicit $z_{0m,b}$ input field from Prigent et al. (2005) may be activated in the revised model version. Similarly, the user

may activate the parameterization of Brock et al. (2006) for $z_{0m,s}$ as a function of accumulated snow melt. Finally, we replaced the parameterization of Zeng and Dickinson (1998) for $z_{0h,g}$ and $z_{0q,g}$ with the parameterization of Yang et al. (2008). Overall, our proposed modifications increase $z_{0m}$ in most areas dominated by vegetation, while $z_{0m}$ is decreased considerably in desert regions.

We then assess the effect of these modifications in CLM offline and CESM land–atmosphere coupled simulations. The

decrease of $z_{0m,g}$ warms the land surface in warm deserts considerably during day and, to a lesser extent, during night. On the other hand, the LST decreases over the cold deserts in particular during the winter season. The impact of the raised $z_{0,v}$ varies diurnally, with a cooling effect during day and a warming effect at night. In CESM, the daytime warming of LST over warm deserts is amplified compared to CLM, associated with a decrease in cloud cover and the resultant increase in incoming solar radiation. Overall, the imposed model modifications reduce biases in the LST DTR compared to MODIS both over warm

deserts, where the DTR is underestimated, and in regions dominated by forests, where the DTR tends to be overestimated. Also, the revisions of $z_0$ alter the local LST response to a conversion of vegetation to bare land considerably, which could be relevant for the simulated biogeophysical effect of desertification. The sensitivity of the LST at 13:30 and the DTR improves in CLM–Z0, while the nighttime sensitivity deteriorates compared to observational data. The response in the TBOT DTR opposes the sign of the LST DTR response, with an amplification in forested regions and a dampening over warm deserts in CESM.

Further, surface wind speeds increase over desert areas, while they decrease in regions with forests. These alterations in surface wind speed typically disappear beyond approximately 1 km above the land surface.

While our revisions of $z_0$ oftentimes improve the simulated LST DTR compared to MODIS, some considerable biases persist, in particular in the case of CESM. Such biases are at least partly related to inadequate properties of the land surface other than $z_0$. For example, the surface emissivity varies considerably across different types of land cover (Jin and Liang, 2006).

Values as low as 0.9 are observed over the Sahara desert, differing strongly from the value of 0.96 for soils in CLM. Jin and Liang (2006) demonstrate that such a change in the emissivity can alter the simulated temperature and surface energy fluxes relevantly. Additionally, several steps are already underway to improve the diurnal variability of temperatures and surface fluxes





over vegetation in CLM. Bonan et al. (2018) replace the big–leaf approach in CLM with a multi–layer canopy and introduce a roughness sublayer parameterization for tall canopies. The latter modification could ultimately replace $z_{0,v}$ entirely. Further,
the addition of biomass heat storage to CLM improved the realism of simulated energy fluxes and LSTs over forests (Swenson et al., 2019; Meier et al., 2019). Some discrepancies between our simulations and MODIS could also be related to the coupling fields that CLM receives, be it from the GSWP3 reanalysis data in the case of the CLM simulations or from the atmospheric component of CESM for the coupled simulations.

We would like to emphasize the value of $z_0$ observations for this work, but also for other efforts of model and parameter-
ization development. Several decades of endeavours to observe $z_0$ allow to better constrain it in models and understand its relation to conditions at the land surface. Yet, knowledge gaps remain in particular for ice sheets. In situ observations indicate that $z_{0m,i}$ varies substantially, likely related to variations in the structure of the ice (Brock et al., 2006; Fitzpatrick et al., 2019). However, the surface structure of the ice is not explicitly simulated in earth system models. Therefore, remote sensing–based data of $z_{0m,i}$ over the ice sheets might be a good solution to capture such spatial variations in $z_{0m,i}$, similar to what already
exists for $z_{0m,b}$. In urban environments, $z_0$ is not only closely linked to mean building height and the density of buildings, but also to the variability of the building height (Nakayama et al., 2011; Kanda et al., 2013). If a global data set of variability of building heights in urban environments becomes available, it could therefore be considered as an additional input variable to compute $z_0$ in the urban module of CLM.

While observations of $z_0$ provide valuable information for model development, the assumptions within the model world
can differ from the assumptions made to estimate $z_0$ in the field. For example, the formulations for the stability correction functions in Hu20 differ from the ones in CLM. Consequently, CLM would produce slightly different turbulent fluxes than measured and used to derive $z_0$ in the field, even if conditions are exactly the same. We would like to highlight that the current approach in CLM of dividing grid cells into tiles of differing land covers does not further specify how the different land covers are situated within this cell. For example, CLM treats a savanna covered by sparse trees and grasses the same as one large
forest next to a grassland landscape (given that the two types of vegetation and the area fraction covered by each vegetation type are roughly the same). But in terms of $z_0$ and other surface properties these two landscapes differ. It might therefore be a consideration to further refine the tile approach in CLM, such that these two landscapes may be distinguished. In CLM, the ecosystem demography model FATES resolves this issue to some extent (Fisher et al., 2015). However, our updates of $z_{0,v}$ after Ra92 are not yet implemented in this version of the model.

Overall, our results highlight the importance of $z_0$ for the exchange of energy, water, and momentum between the land surface and the atmosphere and through that for temperatures at the land surface as well surface wind speed. Beyond these, there are several avenues of impacts we did not explore in this study. For example, we disabled the carbon cycle in our simulations. Thus, we ignore potential consequences for the exchange of greenhouse gases between the land and the atmosphere, be it directly through alterations of the turbulent exchange of such gases or indirectly through biogeophysical effects that affect
biogeochemical processes such as photosynthesis or respiration. Further, the resultant increase in surface wind speed in arid and semi–arid regions are likely to affect mineral dust emissions (Csavina et al., 2014) and might thereby affect existing model biases in CESM (Wu et al., 2019).



*Code and data availability.* The CLM code, the CESM code, the MODIS-based data on the sensitivity of LST to a conversion of vegetation to bare land, and the estimated climatology of the incoming shortwave radiation at the land surface in GSWP3 under clear–sky conditions are

available at https://doi.org/10.3929/ethz-b-000503165. MYD11C3 can be downloaded from https://lpdaac.usgs.gov/products/myd11c3v006/ and Land Cover CCI from http://maps.elie.ucl.ac.be/CCI/viewer/download.php. For the data from Hu et al. (2020) contact Xiaolong Hu and for the data from Prigent et al. (2005) Catherine Prigent. Any model output is available upon request from Ronny Meier.

## Appendix A: Appendix

### A1 Sensitivity tests to isolate contributions from individual modifications

Besides CLM–CTL and CLM–Z0, we run a number of additional simulations to better understand the importance of the individual modifications introduced in CLM–Z0, which are summarized in Table A1. First of all, we run a simulation, CLM–Z0C, that follows the same protocoll as CLM–Z0, but with the median values for $z_{0m,b}$ and $z_{0m,s}$ depicted in Fig. 2 instead of using the spatially explicit data of Prigent et al. (2005) and the parameterization of Brock et al. (2006), respectively. Additionally, we start three 15–year simulations starting from the initial conditions of CLM–CTL that only utilize a subset of the modifications

described in the Section 2. CLM–VEG uses only the parameterization of Raupach (1992) for $z_{0,v}$ but preserves the default for $z_0$ otherwise. In CLM–Z0M, we introduce all the modifications related to $z_{0m}$ but retain the formulation of Zeng and Dickinson (1998) for $z_{0h,g}$ and $z_{0q,g}$. CLM–Ya08 on the other hand applies the formulation of Yang et al. (2008) for $z_{0h,g}$ and $z_{0q,g}$ and uses the default representation of $z_{0m}$. For the latter three simulations we use the years 1998–2002 as an additional spinup period and only analyze 2003–2012.

**Table A1. Overview of CLM simulations.** From left to right, name of simulation, parameterization for $z_{0,v}$, $z_{0m,b}$ $z_{0m,s}$, choice of $z_{0m,i}$, parameterization for $z_{0h,g}$ and $z_{0q,g}$, and initial conditions used. Parameterizations and data sets that are marked with a asterisk were modified before including them in CLM.

| Simulation | $z_{0,v}$ | $z_{0m,b}$ | $z_{0m,s}$ | $z_{0m,i}$ | $z_{0h,g}$, $z_{0q,g}$ | Initial cond. |
|---|---|---|---|---|---|---|
| CLM–CTL | Zeng and Wang (2007) | 0.01 m | 0.0024 m | 0.01 m | Zeng and Dickinson (1998) | 50–year spinup |
| CLM–Z0C | Raupach (1992)* | 0.00085 m | 0.00078 m | 0.0023 m | Yang et al. (2008) | 50–year spinup |
| CLM–Z0 | Raupach (1992)* | Prigent et al. (2005)* | Brock et al. (2006)* | 0.0023 m | Yang et al. (2008) | 50–year spinup |
| CLM–VEG | Raupach (1992)* | 0.01 m | 0.0024 m | 0.01 m | Zeng and Dickinson (1998) | CLM–CTL |
| CLM–Z0M | Raupach (1992)* | Prigent et al. (2005)* | Brock et al. (2006)* | 0.0023 m | Zeng and Dickinson (1998) | CLM–CTL |
| CLM–Ya08 | Zeng and Wang (2007) | 0.01 m | 0.0024 m | 0.01 m | Yang et al. (2008) | CLM–CTL |

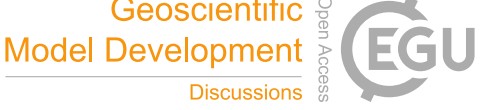

Here, we compare the effect on the annual mean LST DTR of the different sensitivity experiments in comparison to CLM–CTL. The alterations in $z_{0,v}$ alone introduced in CLM–VEG decrease the DTR in regions dominated by forests (where the $z_{0,v}$ is increased) and increase it in regions with a considerable amount of crops (for which $z_{0,v}$ is decreased) compared to CLM–CTL (Fig. A5 b). Interestingly, the response in forested regions is often weaker in CLM–VEG than in CLM–Z0 or even reversed in sign in the Sahel region (Fig. A5 a). The full signal strength only emerges, when the alterations of $z_{0m,g}$ are

introduced in CLM–Z0M (Fig. A5 c). It thus appears that a decrease in $z_{0m,g}$ under a closed canopy dampens the LST DTR. The opposite is the case over warm desert areas. Somewhat unexpected, the amplifications of diurnal variations in LST over arid and semi–arid regions is moderated when Ya08 is introduced in CLM–Z0 compared CLM–Z0M over most of the Sahara, the Middle East, and the Himalaya (Fig. A5 f). On the other hand, the introduction of the Ya08 parameterization for $z_{0h,g}$ and $z_{0q,g}$ with the default $z_{0m,g}$ in CLM–Ya08 enhances the LST DTR (Fig. A5 d). Ya08 therefore amplifies the diurnal LST

variability for relatively large values of $z_{0m,g}$ (which are used in CLM–Ya08 and CLM–CTL), while it dampens this variability for small $z_{0m,g}$ values (which are used in CLM–Z0M and CLM–Z0) compared to the parameterization of Zeng and Dickinson (1998). The globally constant $z_{0m,b}$ in CLM–Z0C is larger than the spatially explicit data in Pr05 (Fig. 2). Also, $z_{0m,s}$ is higher in CLM–Z0C over most regions than in CLM–Z0, with the notable exception of some areas of Greenland (not shown). Thus, $z_{0m,g}$ is generally decreased less in CLM–Z0C than in CLM–Z0 in comparison to CLM–CTL. Accordingly, the response in

the LST DTR tends to be slightly smaller in magnitude in CLM–Z0C than in CLM–Z0 (Fig. A5 a and e). Overall, there is however no major difference between CLM–Z0C and CLM–Z0.

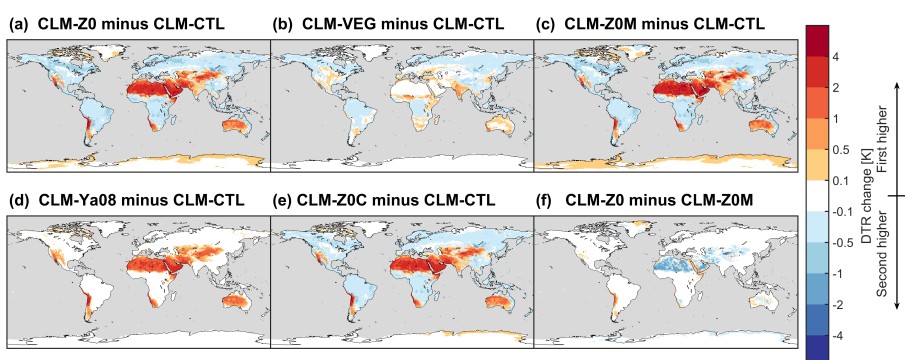

**Figure A1.** As Fig. 8 (a) but over 2003–2012 for (a) CLM–Z0 minus CLM–CTL, (b) CLM–VEG minus CLM–CTL, (c) CLM–Z0M minus CLM–CTL, (d) CLM–Ya08 minus CLM–CTL, (e) CLM–Z0C minus CLM–CTL, and (f) CLM–Z0 minus CLM–Z0M.





## A2 Energy balance decomposition

In this section we present an energy balance decomposition after Luyssaert et al. (2014) to better understand the contribution of changes in individual energy fluxes to the overall change in LST between CLM/CESM–CTL and CLM/CESM–Z0. Assuming
the emissivity of the land surface is equal to one, the change in LST ($\Delta LST$) is expressed as follows:

$$\Delta LST = \frac{1}{4\sigma LST^3} \left( -SW_{in}\Delta\alpha + (1-\alpha)\Delta SW_{in} + \Delta LW_{in} - \Delta LH - \Delta SH - \Delta G - \Delta I \right), \tag{A1}$$

where $\sigma$ is the Stefan–Boltzmann constant, $SW_{in}$ the incoming shortwave radiation, $\alpha$ the albedo, $LW_{in}$ the incoming long-wave radiation, $LH$ the latent heat flux, $SH$ the sensible heat flux, $G$ the ground heat flux, and $I$ the energy imbalance. $\Delta X$ corresponds to the difference in variable $X$ between CLM/CESM–Z0 and CLM/CESM–CTL. We take the average of
CLM/CESM–Z0 and CLM/CESM–CTL for the variables for which no difference is taken between these two simulations (e.g., $SW_{in}$ for the first term in the brackets). The terms on the right hand side of Eq. A1 correspond to the change in LST due to the change in albedo, incoming shortwave radiation, incoming longwave radiation, latent heat, sensible heat, ground heat, and the energy imbalance from left to right.

    Fig. A2 shows the most important terms of the energy balance decomposition at 13:30 during boreal summer in the offline
simulations. Changes in LST during the day between CLM–CTL and CLM–Z0 are mostly the result of alterations in $SH$. The contribution from $SH$ is most of the time compensated partly by $G$. For example, if the LST increases due to a reduction in $SH$ part of this energy surplus is compensated by the energy stored in the ground (leading to a warming of the soils below the land surface). The other terms provide only little to the overall change in LST. At 01:30, $\Delta LST$ is again driven by changes in $SH$ in the high–latitudes (Fig. A2). At lower latitudes, in particular in the warm deserts, the strong LST response during
the day frequently translates into the night through the energy stored in the ground. Over the Sahara, for example, the ground absorbs more energy during the day because $SH$ is reduced, resulting in warmer ground surface temperatures at night.

    For the land–atmosphere coupled simulations, the incoming shortwave and longwave radiation terms become relevant due to atmospheric feedbacks. During boreal summer, increased incoming solar radiation over the Sahara, the Middle East and Himalaya amplify the warming from the reduced $SH$ (Fig. A4). The reduction in LST over the northern mid- and high–latitudes
mostly coincides with less incoming solar radiation. In contrast, the signal in winter is determined by the longwave radiation in those regions (Fig. A5). A warming of atmospheric temperatures over most of the Asian continent and the northern part of North America in CESM–Z0 (Fig. A6 a) causes in increase in the incoming longwave radiation, which induces a warming of the LST in those regions.



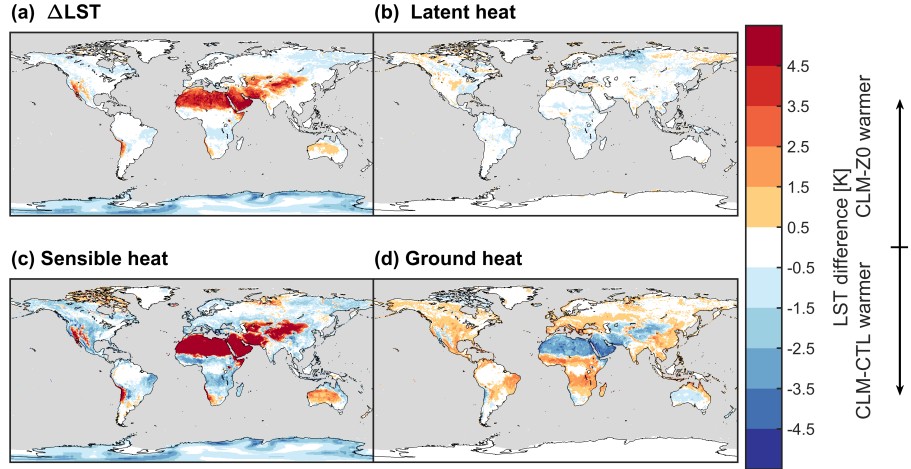

**Figure A2. Energy balance decomposition for change in LST at 13:30 local solar time in boreal summer of CLM–Z0 minus CLM–CTL.** Panel (a) change in LST, (b) contribution from change in latent heat, (c) contribution from change in sensible heat, and (d) contribution from change in ground heat flux. Note that some terms are not shown because they are zero in offline simulations (incoming radiation terms) or because they are small (albedo, and imbalance term).

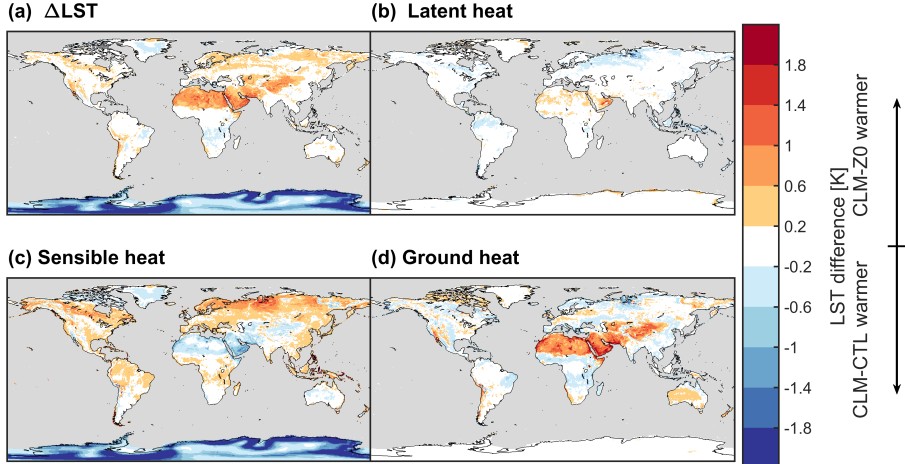

**Figure A3.** As Fig. A2 but at **01:30** local solar time.





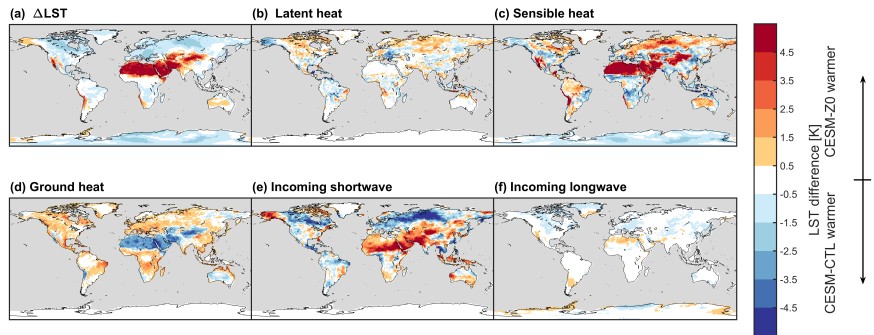

**Figure A4. Energy balance decomposition for change in LST at 13:30 local solar time in boreal summer of CESM–Z0 minus CESM–CTL.** Panel (a) change in LST, (b) contribution from change in latent heat, (c) contribution from change in sensible heat, (d) contribution from change in ground heat flux, (e) contribution from change in incoming shortwave radiation, and (f) contribution from change in incoming longwave radiation. Note that the albedo and the imbalance term are not shown because they are small.

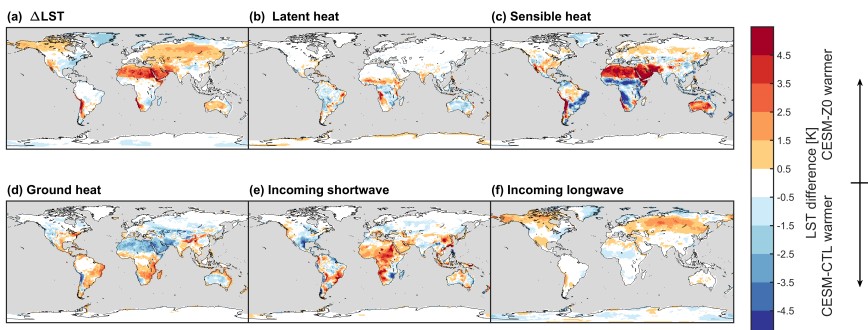

**Figure A5.** As Fig. A4 but for boreal winter.

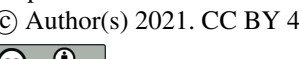


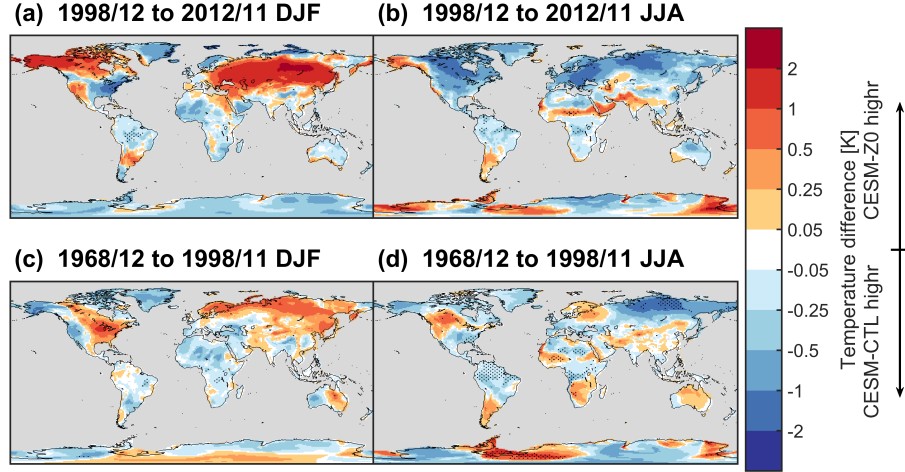

**Figure A6.** As Fig. 12 (a) and (b) but using data of the analysis period only (top row, December 1998 to November 2012) and using data from the last 30 years of the spinup period (bottom row, December 1968 to November 1998).

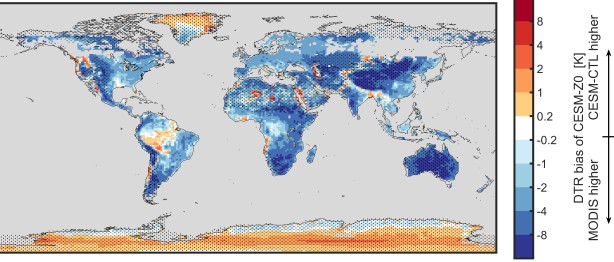

**Figure A7.** As Fig. 11 (d) but for CESM–Z0.

*Author contributions.* RM and ELD conceptualized the study with help from GBB, DL, and SIS. RM implemented the modifications in the
model, with the help of ELD, DL, and GBB. XH provided the vegetation surface roughness observations and helped with their analysis. GD conducted the analysis for the sensitivity of land surface temperatures to desertification. CP provided the data from Prigent et al. (2005). RM conducted the analysis and drafted the manuscript with help from all co-authors. All authors contributed to the interpretation of the results and preparation of the manuscript.

*Competing interests.* The authors declare no competing interests.



*Acknowledgements.* The CESM project is supported primarily by the National Science Foundation. We thank all the scientists, software engineers, and administrators who contributed to the development of CESM2. RM was funded by the Swiss National Science Foundations (SNSF; http://p3.snf.ch/Project-172715; grant no. 200021_172715). We thank Wim Thiery for his valuable advice. All graphics were created with MATLAB R2020b.





**Table A1.** List of abbreviations and symbols used in this study. Symbols that only appear in one equation are not listed.

| Abbreviation | Long name/description |
| --- | --- |
| $c$ | Empirical constant in Ra92 [ ] |
| $c_{d1}$ | Constant in Ra92 (= 7.5) [ ] |
| CESM | Community Earth System Model (version 2.1.2) |
| CLM | Community Land Model (version 5.1) |
| $C_R$ | Drag coefficient of an isolated roughness element [ ] |
| $C_S$ | Drag coefficient of the ground in the absence of vegetation [ ] |
| $d$ | Displacement height [m] |
| DTR | Diurnal temperature range |
| Du18 | Potential change in LST for a conversion of vegetation to bare land after Duveiller et al. (2018) [K] |
| $G$ | Ground heat flux [$\mathrm{W\,m^{-2}}$] |
| GSWP3 | Global Soil Wetness Project reanalysis product version 3 |
| $h_{top}$ | Canopy height [m] |
| Hu20 | $z_{0,v}$ observations of Hu et al. (2020) |
| $LAI$ | Exposed leaf area index [$\mathrm{m^2\,m^{-2}}$] |
| LISSS | Lake, Ice, Snow, and Sediment Simulator (Lake model in CLM) |
| LST | Land surface temperature [K] |
| $M_a$ | Accumulated snow melt [m w.eq.] |
| MODIS | Moderate resolution imaging spectroradiometer |
| MYD11C3 | Monthly MODIS LST product (version 6) |
| PFT | Plant functional type |
| Pr05 | $z_{0m,b}$ data of Prigent et al. (2005) [m] |
| $SAI$ | Exposed stem and dry leaf area index [$\mathrm{m^2\,m^{-2}}$] |
| $SH$ | Sensible heat flux [$\mathrm{W\,m^{-2}}$] |
| TBOT | Temperature at the bottom of the atmospheric column [K] |
| $u_*$ | Friction velocity [$\mathrm{m\,s^{-1}}$] |
| $V$ | Fractional weight for $z_{0,v}$ between vegetation and $z_{0m,g}$ [ ] |
| $VAI$ | Vegetation area index = $LAI + SAI$ [$\mathrm{m^2\,m^{-2}}$] |
| $VAI_{off}$ | Offset of $VAI$ [$\mathrm{m^2\,m^{-2}}$] |
| $W_S^{cs}$ | Climatology of the incoming solar radiation at the surface [$\mathrm{W\,m^{-2}}$] |
| $W_{TOA}$ | Theoretical daily incoming solar radiation at the top of the atmosphere according to Berger (1978) [$\mathrm{W\,m^{-2}}$] |
| Ra92 | $z_{0,v}$ parameterization after Raupach (1992) and Raupach (1994) |
| Ya08 | Parameterization of $z_{0h,g}$ and $z_{0q,g}$ after Yang et al. (2008) |
| $z_0$ | Surface roughness [m] |
| $z_{0h}$ | Surface roughness for sensible heat [m] |
| $z_{0m}$ | Momentum (aerodynamic) surface roughness [m] |
| $z_{0q}$ | Surface roughness for latent heat [m] |
| $z_{0,b}$ | Surface roughness of bare soil (with additional subscripts h, m, or q) [m] |
| $z_{0,g}$ | Surface roughness of the ground (with additional subscripts h, m, or q) [m] |
| $z_{0,i}$ | Surface roughness of ice and glaciers (with additional subscripts h, m, or q) [m] |
| $z_{0,v}$ | Aerodynamic surface roughness for exchange between canopy air space and atmosphere [m] |
| $z_{0,s}$ | Surface roughness of snow (with additional subscripts h, m, or q) [m] |
| $\lambda$ | Roughness density of vegetation [ ] |
| $\kappa$ | von Karman constant (= 0.4) [ ] |
| $\nu$ | Kinematic viscosity of air (= 1.5e-5$\mathrm{m^2\,s^{-1}}$) |





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
