# Peer review of "Impacts of a Revised Surface Roughness Parameterization in the Community Land Model 5.1"

_Geoscientific Model Development, 2021_

## Author Comment (AC1)

We would like to thank the reviewer for the constructive comments that helped to make the manuscript more understandable. In the following, we list the reviewers' comments in black and our reply in blue. When appropriate, we show the revised part of the manuscript in green in this response letter. Note that references to locations in the manuscript correspond to the marked-up version of the manuscript, if not stated otherwise.

The most important changes include the following:

-We have revised the parameterization of the vegetation surface roughness, which led to the elimination of the parameter $VAI_{off}$. Accordingly, the vegetation surface roughness in the current model revision differs from the previous revision, which slightly affects all of the presented results, but without altering the main conclusions.

-The formulation of the text was improved in particular in Sections 2.2, 2.3, 4.1, 5.2, and 6.

-We decided to add the cloud masking in panels (a) and (d) of Figs. 8 and 11 for consistency within these figures.

*CLM does not explicitly compute and output the LST. It is necessary to give the equation for the LST computation in text.*

We have added a description of LST computation in CLM in the Section 4.2 (L455-463).

*The processing MODIS LST in section 4.1 is stated in texts and is not easy to be captured. It suggests giving the bullet-point of each procedure as these in section 4.2.*

We agree that this section was difficult to follow. We do not think however that a bullet-point list is the appropriate measure to solve this, as some steps require more background information, are parallel rather than sequential to other steps, and/or are inseparably connected to other steps. Therefore, we decided to rewrite and restructure this section to make it more understandable (L433-453). If the reviewer would like to read a more detailed description of the method, we kindly refer the reviewer to previous studies that employed the same methodology (Duveiller et al., 2018 and Duveiller et al., 2021).

*In both L488-489 and L491-493, the relationships between incoming shortwave radiation and LST seems oppositely.*

Thank you, this was a mistake. This sentence now reads (L572-574):

Over the northern mid- and high–latitudes, an **increase** in cloud cover during summer coincides with a reduction in daytime LSTs due to less incoming shortwave radiation (Figs. A4).

*From the computation equation of LST (e.g., Yang et al. 2008 in reference list; Wang et al. 2014), the long-wave radiations (both incoming and outgoing) directly determine the LST magnitude. The incoming longwave radiation may change due to the cloud cover. The incoming shortwave radiation does not directly influence LST. Therefore, it is necessary to explain the possible mechanisms behind the relationship between incoming shortwave radiation and LST/cloud cover.*

The calculation of the LST in CLM5.0 has been revised in comparison to CLM4.0, as is now described in the text thanks to a previous comment of the reviewer (L455-463). The current computation of the LST is linked to the temperatures of the leaves and ground and to their respective energy balances and not only to the longwave radiation fluxes. Accordingly, the resultant LST is affected by all energy fluxes rather than just the longwave radiation. We hope that the description of the LST computation in our simulations clarifies this. Also, we would like to thank the reviewer for making us aware of the study from Wang et al. (2014), which supports our revisions of the ground surface roughness and our results. We cited this study where appropriate (L69, L80).

References

Duveiller, G., Hooker, J., and Cescatti, A.: The mark of vegetation change on Earth's surface energy balance, Nat. Commun., 9,https://doi.org/10.5194/essd-2018-24, 2018.

Duveiller, G., Filipponi, F., Ceglar, A., et al.: Revealing the widespread potential of forests to increase low level cloud cover, Nat. Commun.,12, https://doi.org/https://doi.org/10.1038/s41467-021-24551-5, 2021

---

## Author Comment (AC2)

We would like to thank the reviewer for the constructive comments that help to make the manuscript more understandable. In the following, we list the reviewers' comments in black and our reply in blue. When appropriate, we show the revised part of the manuscript in green in this response letter. Note that references to locations in the manuscript correspond to the marked-up version of the manuscript, if not stated otherwise.

The most important changes include the following:

-We have revised the parameterization of the vegetation surface roughness, which led to the elimination of the parameter $VAI_{off}$. Accordingly, the vegetation surface roughness in the current model revision differs from the previous revision, which slightly affects all of the presented results, but without altering the main conclusions.

-The formulation of the text was improved in particular in Sections 2.2, 2.3, 4.1, 5.2, and 6.

-We decided to add the cloud masking in panels (a) and (d) of Figs. 8 and 11 for consistency within these figures.

*I found this article very difficult to read as several sections were unpolished and need copy-editing. I wish that I could have spent more time concentrating on the science than trying to understand what the authors really meant. Several typos or wrong references to Figures made me think that the co-authors had not thorough y proof-read othe manuscript. This could be a requirement before an editor passes on papers to reviewers. More specifically, paragraph 2.2 which has been written hastily and needs a complete rewrite.*

We apologize if it was difficult to follow the manuscript. Several sections were revised according to the reviewer's comments as well as to the comments of reviewer 1 (e.g. Section 2.2). We hope the study is in an adequate form now.

*Line 328 the aurhors refer Fig 4b as illustrating forests when Fig. 4b is for grasslands, you probably meant Fig 4c which is for forests. Similarly on line 331 you refer to "Fig. 4c and d" for grassland and crops, I believe that you meant "Fig 4b and d".*

We reordered the labels of Figs. 4 and 11 shortly before submission to make the labels consistent throughout the manuscript. Unfortunately, we forgot to adjust some of the references in the text accordingly. This is corrected now (L381, L385).

*I recommend that this paper undergoes copy-editing before it is published as in its present form few readers will read it to the end.*

Several authors carefully read through the manuscript again, including a native English speaker. We believe this should be sufficient.

*Line 59: dust does not necessarily cool surface temperatures, it can also warm them. Whether dust has a warming or a cooling effect on surface temperature depends on the surface albedo, the size of the particle and the mineralogy of dust (see Liao et al, 1999; Claquin et al 1998).*

The respective sentence has been adjusted and the references of Claquin et al. (1998) added (L70-72):

The aerodynamic $z_0$ also affects the simulated mineral dust emissions (Menut et al., 2013), which absorb and reflect solar radiation and **thereby alter** temperatures at the land surface (**Claquin et al., 1998**; Miller and Tegen, 1998; Klose et al.,2021).

*Line 96: typo, change 'cylce' to 'cycle'*

Corrected (L112).

*Lines 144-145: you say that Fig 1f opposes observations. I could not find the lines representing the observations in Fig 1f. Overall, Figure 1 should be improved, the colored lines are too faint and should be well referenced.*

This sentence was reformulated to avoid any ambiguity (L199-201):

Further, CLM produces low values of $z_{0,v}$ in the absence of leaves for broadleaf deciduous forests, resulting in an annual cycle of $z_{0,v}$ that is in contradiction to Hu20 (Fig. 1 f) and other observational studies (Nakai et al., 2008; Maurer et al., 2013; Young et al., 2021).

Also the width of the lines shown in Fig. 1 was increased and the reader is explicitly directed to the appropriate figure elements at several places in the text (L152-153, L162, L199, and L239).

*Line 196: change "For cw  and V Aloff  we use a precision of 0.1, for CR  and c  0.01, and for CS  0.001. " to "The precision used for these key parameters is respectively: 0.1 for cw, 0.01 for CR  and c  and 0.001 for CS."*

We now show the tested range as well as the precision in Table 1 as indicated in the text at L236-237. Also note that the optimization of Ra92 has changed due to another comment of the reviewer, as described in more detail below.

*Lines 197-198: "Overall, the optimized Ra92 parameterizations improve the mean seasonal cycle of z0,v  for all vegetation types (right column Fig. 1)." Indicate clearly that the line representing this seasonal cycle is the solid turquoise line so that the reader can readily grasp which lines to compare CLM with.*

This is now clarified in the brackets (L240-241):

The optimized Ra92 parameterizations improve the mean annual cycle of $z_{0,v}$ for all vegetation types (**compare orange to red lines in right column Fig. 1 in reference to turquoise lines**).

*I did not find the definition of cw in the text and it should be added to Table 1.*

$c_w$ is now defined in the text (L214) and was included in Table A1.

*For your set of 5 parameters (cw Cs CR c and V Aloff), did you check that the set that constitutes your solution is unique? Could there be multiple sets of solutions? A Latin Hypercube method can find these multiple solutions. lmax seems to be missing from Table 1*

Motivated by this comment, we have revised the fitting procedure of Ra92. Specifically, we were able to remove the $VAI_{off}$ parameter entirely without deteriorating the fits. This is preferable as the employed parameterization is now closer to the one originally proposed in Raupach (1992) and Raupach (1994) and it reduces the number of input parameters for the model. The fitting procedure is now described in more detail in the text (L231-238).

Note that the optimal solution over the given range and at the given precision is unique for each vegetation type. We think that a Latin Hypercube might miss an optimal solution, as the different parameters interact in a non-linear fashion. Also, such an approach does not appear necessary, since we are able to test all possible parameter combinations over the desired range and at the desired precision by brute force.

*Line 249. I would be careful when using the terminology 'an ideal solution'. How do you define what an ideal solution is?*

This sentence was rephrased to (L298-300):

Overall, there is clear evidence that the parameterization of $z_{0h,b}$ and most likely also $z_{0q,b}$ applied currently in CLM5 is in disagreement with observations in the field.

*Line 500 mentions a comparison of diurnal temperature range (DTR) with MODIS observations, I could not find the MODIS observation of Figure 11b, did you mean Fig 11d?*

Again, we forgot to update the label in the text. This is now correct (L583-584).

*ll 533-534: replace 'reacts comparably strong' with 'shows a response stronger by a factor 3'*

We think the formulation proposed by the reviewer would change the meaning of the sentence. Since this statement appears to be misleading, it was reformulated (L617-620):

At grid cell level, the LST DTR **exhibits a similar dependence on the change in $z_{0m}$ between CESM–Z0 and CESM–CTL, if the $z_{0m}$ changes by no more than a factor of 3,** as visible by values between -0.5 to 0.5 on the x–axis in Fig. 13 b.

*Line 542: replace 'And forth' with 'And fourth'*

Corrected (L631).

*Pp 26-27 needs re-writing.*

This section was revised (L614-640). We hope it is now possible to follow the content of the text.